# A Radiative-Convective Model based on constrained Maximum Entropy Production

Vincent Labarre[1,2], Didier Paillard[1], and Bérengère Dubrulle[2]

[1]Laboratoire des Sciences du Climat et de l'Environnement, CEA Saclay, Orme des Merisiers, Gif-sur-Yvette, France
[2]SPHYNX/SPEC/DSM, CEA Saclay, Orme des Merisiers, Gif-sur-Yvette, France

**Correspondence:** Vincent Labarre (vincent.labarre@lsce.ipsl.fr) and Didier Paillard (didier.paillard@lsce.ipsl.fr)

**Abstract.** The representation of atmospheric convection induced by radiative forcing is a longstanding question mainly because turbulence plays a key role in the transport of energy as sensible heat, geopotential, and latent heat. Recent works have tried using the Maximum Entropy Production conjecture as a closure hypothesis in 1D Simple Climate Models to compute implicitly temperatures and the vertical energy flux. However, these models fail to reproduce realistic profiles. To solve the problem, we describe the energy fluxes as a product of a positive mass mixing coefficient with the corresponding energy gradient. This appears as a constraint which imposes the direction and/or limits the amplitude of the energy fluxes. It leads to a different MEP steady state which naturally depends on the considered energy terms in the model. Accounting for this additional constraint improves the results. Temperature and energy flux are closer to observations, and we reproduce stratification when we consider the geopotential. Variations of the atmospheric composition, like a doubling of the carbon dioxide concentration, are also investigated.

## 1 Introduction

The climate system is complex and usually divided into different components: atmosphere, ocean, cryosphere, lithosphere and biosphere (Peixoto and Oort, 1992).There are different approaches to climate modeling (Randall et al., 2007). We can classify them in three main classes. Global Climate Models (GCMs) are the more sophisticated ones (see Dufresne et al. (2013) for an example). They explicitly represent the circulation of the atmosphere and ocean. Earth Models of Intermediate Complexity (EMICs) simulate the Earth system with more simplifications than GCMs (see Goosse et al. (2010) for an example). These simplifications allow simulations over larger time periods, which is useful to study past climates. Simple Climate Models (SCMs) use only a few key processes to answer specific questions (see Paillard (1998) for an example).

Both the complex and simple models have different strengths and weaknesses and are used for different applications. For example, GCMs are largely used to make climate projections for the next century. Since the numerical resolution of dynamical equations from the micro-scale (of order $\simeq 10^{-3}$ m for viscous dissipation) to the scale of interest ($\simeq 10^7$ m for the typical

size of the Earth) is still impossible, GCMs, however, need to represent sub-grid processes such as small eddies, convection or cloud's formation. To do it, models usually express the intensity of fluxes due to unresolved phenomena as a function of the resolved-variables. This approach is called a "turbulent closure" and usually requires the introduction of empirical parameters such as turbulent master length scale, turbulent velocity diffusion terms. (Mellor and Yamada, 1982), or the use of different

quantities like Convective Available Potential Energy (CAPE) or Convective Inhibition (CIN) to fix the convective intensity (Yano et al., 2013). These parametrizations change from one model to another, resulting in different predictions (Stevens and Bony, 2013). They also require adjusting the numerous free parameters ("tuning") in order to track observations (Hourdin et al., 2017). SCMs appear as an interesting alternative to when considering past or future climate study over a larger period like glacial-interglacial cycles. Indeed, an SCM is a set of a reasonable number of equations and physical quantities, providing

an easier assessment of the impact of the parameters of the models (like the concentration of greenhouse gases for atmospheric models). Many SCMs are based on the idea that the computation of all the microscopic details may be unnecessary if we are interested in quantities at larger spatiotemporal scales.

Consequently, a lot of SCMs describe the Earth simply with energetic considerations (North et al., 1981). Those models are called Energy Balance Models (EBMs). The atmosphere is mainly driven by radiative forcing: solar radiations give energy to

the Earth which emits infra-red radiations to Space. This heating is not homogeneous around the globe for different reasons like geometry or variations of albedo with the nature of the ground. The insolation is more important for tropics than poles and leads to latitudinal heat transport. For the vertical axis, the amount of radiative energy absorbed by the Earth naturally depends on the atmospheric components. The ground usually receives more solar radiations because of the relative transparency of the atmosphere. As a result, the atmosphere is heated from below which may lead to unstable situations where the temperature gra-

dient exceeds the adiabatic gradient. Then, this causes atmospheric motion and vertical heat transport named convection.Since EBMs are usually based only on the energy budget, it is necessary to specify a relation between the energy fluxes and the temperature gradient, called a closure hypothesis. So EBMs mainly differ by the representation of the fluxes between the fluids layers of the Earth (atmosphere and ocean). For example, horizontal fluxes are sometimes represented by purely diffusion terms (North et al., 1981). On the other hand, vertical energy transport have been modeled using different approaches like the

convective adjustment. The latter consists in computing the temperature profile at radiative equilibrium for stable regions and adjusting it where the critical gradient is exceeded (Manabe and Strickler, 1964). Representing both the horizontal and the vertical energy fluxes is an important issue for EBMs. This concerns the subject of the present paper.

Since the seventies (Paltridge, 1975), Maximum Entropy Production (Martyushev and Seleznev, 2006) (MEP) is also used as a closure hypothesis in EBMs. This conjecture stipulates that the climatic system (or one of its component) optimizes its

entropy production due to internal heat transfers. It allows computing implicitly (i.e. without the computation of the dynamics) horizontal fluxes (O'brien and Stephens, 1995; Lorenz et al., 2001), and vertical fluxes (Ozawa and Ohmura, 1997; Pujol and Fort, 2002) without the parametrizations required in more conventional models. Former MEP based Models (MEPMs) have been criticized for three main reasons. One is the absence of dynamics and the validity of MEP (Rodgers, 1976). The second criticism deals with the extra parametrizations or the assumptions used in MEPMs. Indeed, one may ask ourself if the suc-

cesses of the models are really due to the MEP hypothesis, or to tuning or other ingredients (Goody, 2007). The final criticism

concerns the usually simplified description of the radiative forcing in these models. Recently, a MEPM overcoming the last two criticisms has been built in Herbert (2012). It includes a refined description of the radiative budget in the Net Exchange Formalism, without extra assumptions. The only adjustable quantities concern the radiative budget, such as the albedo, and not the atmospheric or oceanic energy transport. The model provides a relatively good approximation for the temperature and
horizontal heat fluxes (Herbert et al., 2011b).

However, the vertical energy fluxes are still overestimated in such models when compared to observations or conventional Radiative-Convective Models (RCMs) like Manabe and Strickler (1964). Furthermore, the energy fluxes are not always oriented against the energy gradient and it does not predict stratification in the upper atmosphere. This is not surprising because geopotential wasn't taken into account. Yet, we know from fluid mechanics that gravity plays a major role in natural convection
(Rieutord, 2015). Gravity is also obviously responsible for stratification in the upper atmosphere. In this paper, we develop a MEPM that describes more properly the atmospheric convection. In the same spirit as the previous SCMs, we do not attempt to resolve the dynamical equations, but we add only some keys features. Two ingredients are introduced to represent vertical heat fluxes more correctly. The first one is to describe energy transport as the product of a nonhomogeneous mixing mass coefficient, times the specific energy gradient. This brings a new constraint into a MEPM. The second one is to consider different energy
terms: sensible heat, geo-potential and latent heat. We show that this simplified description of the energy transport, combined with MEP closure hypothesis, can lead to relatively realistic results.

The outline of this paper is as follows. In the first part, we describe our model, presenting the transport of heat by mixing. The formulation of the constrained MEP optimization problem is given (part 2). Then, we compute the temperature, the specific energy, and the energy fluxes profiles. We give a physical interpretation of the effect of the constraint emerging from the posi-
tivity of the mass mixing coefficient. The impact of different expressions for energy is discussed (part 3). A sensitivity test for the concentration of $O_3$ and $CO_2$ is also performed. Finally, we discuss further works and objectives (part 4). The computation of the geopotential is given in annex A, and the resolution of the optimization problem is described in annex B.

## 2  Model

### 25  2.1  Vertical structure of the atmosphere

The atmosphere is divided into a column of $N$ vertical layers. We work with prescribed pressure levels, so the elevation $z$ depends on the temperature profile (see annex A). The $CO_2$, $O_3$ and water vapor profiles are fixed according to observations of A. McClatchey et al. (1972) and the ground is represented by a layer with a fixed surface albedo $\alpha$. The atmosphere is supposed to be in hydrostatic equilibrium and is considered as an ideal gas. The specific energy (energy per unit mass) in layer
$i$, of mean elevation $z_i$, temperature $T_i$ and mixing ratio $q_i$ (ratio between the mass of water vapor and total mass of the air for a given volume) is the so-called moist static energy

$$e_i = C_p T_i + g z_i + L q_i, \tag{1}$$

where $C_p = 1005$ J.kg$^{-1}$.K$^{-1}$ is the heat capacity of the air, $g = 9.81$ m.s$^{-2}$ is the terrestrial acceleration of gravity and $L = 2,5.10^6$ J.kg$^{-1}$ is the latent heat of vaporization.

We note $\mathcal{R}_i$ the net radiative energy input in layer $i$ taking into account several effects: shortwave radiation, longwave radiation, reflexion, and reabsorption. More explicitly:

$$\mathcal{R}_i = SW_i + LW_i = SW_{i\downarrow} - SW_{i\uparrow} + LW_{i\downarrow} - LW_{i\uparrow}$$

where $SW_{i\downarrow}$ is the downward radiative energy flux for shortwaves, $SW_{i\uparrow}$ is the upward radiative energy flux for shortwaves, $LW_{i\downarrow}$ is the downward radiative energy flux for long waves, and $LW_{i\uparrow}$ is the upward radiative energy flux for long waves.

We use the code developed in Herbert et al. (2013) to compute the radiative budget. This model was developed to give a realistic description of the absorption properties of the more radiatively active constituents of the atmosphere while keeping a smooth dependence of the radiative flux with respect to the temperature profile. As suggested by the authors, this last requirement is important in the framework of a variational problem. The model is based on Net Exchange Formalism (Dufresne et al., 2005), where the basic variables are the net exchange rates between each pair of layers instead of radiative fluxes.

In the longwave domain, the code decomposes the spectrum into 22 narrow bands, and in each band, it accounts for absorption by water vapor and carbon dioxide only. The absorption coefficient is computed using the statistical model of Goody (1952) with the data from Rodgers and Walshaw (1966). For the spatial integration, the diffusive approximation is performed with the standard diffusion factor $\mu = 1/1.66$. Apart from the absorption data, given once and for all, the inputs of the model are the water vapor density, temperature profile and carbon dioxide concentration. One may either fix absolute or relative humidity. In the shortwave domain, absorption by water vapor and ozone is accounted for by adapting the parameterization from Lacis and Hansen (1974). The input parameters for the model are the water vapor density and ozone density profiles, as well as surface albedo and solar constant. Clouds are not considered in the model. More details can be found in Herbert et al. (2013) and its supplementary material. The net radiative budget for the atmospheric layer $i$, $\mathcal{R}_i$, is given by summing over all terms involving the layer in question. In particular, $\mathcal{R}_i$ is a function of all temperatures $\{T_j\}_{j=0,...,N}$ in the profile. Then,

$$\mathcal{R}_i(T, q, O_3, CO_2, \alpha) = SW_i(q, O_3, \alpha) + LW_i(T, q, CO_2). \tag{2}$$

In the previous equation and in the following $T$, $q$, $O_3$, $CO_2$ will refer to complete profiles (i.e $T = \{T_i\}_{i=0,...,N}$ etc). Given that $q$ (or $h = q/q_s(T)$ fixed relative humidity), $O_3$, $CO_2$ and $\alpha$ are fixed in our model, we will only indicate the $T$ dependence.

The vertical energy flux is represented by mixing between adjacent layers. Then, the net upward energy flux between layers $i$ and $i - 1$ writes

$$F_i = m_i(e_{i-1} - e_i), \tag{3}$$

where $m_i$ is a mixing coefficient that represents a mass per unit time and per unit surface. It is not (necessarily) homogeneous in all the column. We notice that $m_i$ is typically the kind of coefficient that requires, at some point, a parametrization in usual

climate models. Here $m$ is obtained from the MEP procedure.

Taking into account the net radiative energy budget $\mathcal{R}_i$, the energy balance at the stationary state for the layer $i$ reads

$$F_i - F_{i+1} + \mathcal{R}_i = 0, \tag{4}$$

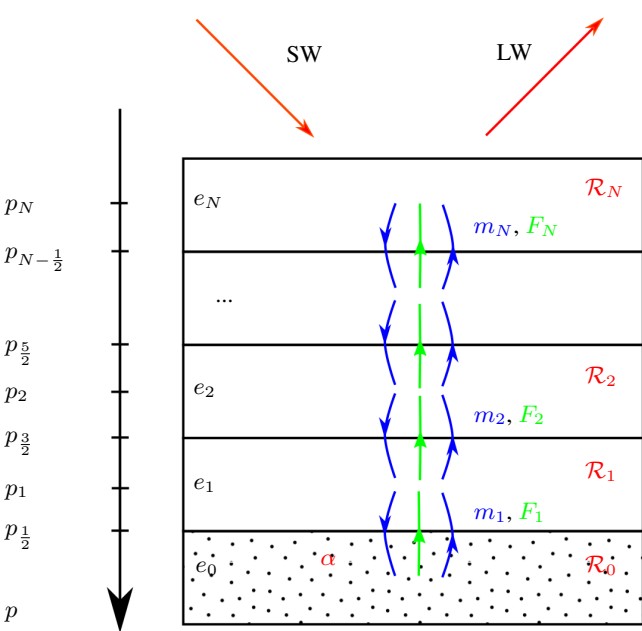

**Figure 1.** Discretization of an atmospheric column into $N$ layers. The layer $i$, at temperature $T_i$, fixed pressure $P_i$, elevation $z_i$ (which depends on the the temperature profile) and mixing ratio $q_i$, has a specific energy $e_i = C_p T_i + gz_i + Lq_i$. $m_i$ is the mass mixing coefficient between layers $i - 1$ and $i$ which leads to the net energy flux $F_i = m_i(e_{i-1} - e_i)$. $\mathcal{R}_i$ is the net radiative energy budget in the layer $i$. The ground is represented by layer 0 with fixed surface albedo $\alpha$.

## 2.2 Maximum Entropy Production with constraint

The principle of our model is to determine the fluxes $F$ and temperatures $T$ with the maximization of the entropy production. In the thermodynamics of diffusive processes, the entropy production is expressed as the sum of products of the fluxes with their associated thermodynamic forces. But we aim here at representing convection, not diffusion. We need to represent how the air parcels are mixed but only after (turbulent) convective motions. We can usefully separate this process in two steps. First, we assume a pseudo-adiabatic motion of an air parcel from one layer to another due to convection. This step is purely

mechanical, without entropy production since we neglect viscous dissipation. The energy of the air parcel is conserved but its temperature and composition may change during the motion. If the air parcel was initially in layer $i$ and goes to the layer $i+1$, the temperature of the air parcel becomes, by conservation of energy,

$$T'_i = T_i + \frac{1}{C_p}\left[g(z_i - z_{i+1}) + L(q_i - q_{i+1})\right]. \tag{5}$$

Here, we have assumed that the water vapour concentration changes pseudo-adiabatically during the convection and not due to the mixing. This is not fully consistent since we do not impose water conservation. Secondly, the air parcel is mixed by diffusion with the ambient air in layer $i+1$ and transfers an amount of sensible heat per unit mass $C_p T'_i$. At the same time, air parcels leave the layer $i+1$ for the layer $i$ with a sensible heat per unit mass $C_p T_{i+1}$. So the net flux of sensible heat due to this process is

$$m_{i+1} C_p (T'_i - T_{i+1}) = m_{i+1}(e_i - e_{i+1}) = F_{i+1}, \tag{6}$$

where $m_{i+1}$ is the mass mixing coefficient between layers $i$ and $i+1$. So the entropy production that results due to the sensible heat exchange between layers $i$ and $i+1$ is the product of the flux $F_{i+1}$ with the thermodynamic force associated to the sensible heat only (i.e. the gradient of inverse temperature)

$$\sigma_{i+1} = F_{i+1}\left(\frac{1}{T_{i+1}} - \frac{1}{T_i}\right). \tag{7}$$

By summing over all layers, and using the fact that $F_{N+1} = 0$, we show that the total entropy production can be written

$$\sigma = \sum_{i=0}^{N} \frac{(F_i - F_{i+1})}{T_i}. \tag{8}$$

In thermodynamics, more terms may contribute to entropy production such as volume work or mixing. As for others MEPM (Kleidon, 2010), we only retain the sensible heat exchange term. The geopotential and latent heat terms appear in the entropy production only as a result of our representation of convective transport, which is supposed to occur as a mechanically induced
mass transport without entropy production. We can easily express the entropy production with temperatures $\sigma(T)$ by using the energy balance in stationary state $F_i - F_{i+1} + \mathcal{R}_i(T) = 0$. Then the problem is usually solved in term of temperature with a global constraint of energy conservation:

$$\left\{ \max_{T_0, \dots, T_N} \left(-\sum_{i=0}^{N} \frac{\mathcal{R}_i(T)}{T_i}\right) \quad \middle| \quad \sum_{i=0}^{N} \mathcal{R}_i(T) = 0 \right\}. \tag{9}$$

Given the form of the energy transport (3), we here need to have additional constraints. Namely, the mass mixing coefficients
$m_i$ must be positive. It is then natural to solve the problem in term of flux by expressing the entropy production $\sigma(F)$ and inequality constraints $m_i \geq 0$ with energy fluxes. Assuming that the relation $\mathcal{R}(T)$ is invertible (annexe B), we can formally write

$$F_{i+1} - F_i = \mathcal{R}_i(T) \qquad \Leftrightarrow \qquad T_i = \mathcal{R}_i^{-1}(F). \tag{10}$$

This results in the following optimisation problem with inequality constraints:

$$\left\{ \max_{F_1,\dots,F_N} \left( \sum_{i=0}^{N} \frac{F_i - F_{i+1}}{\mathcal{R}_i^{-1}(F)} \right) \quad \Big| \quad \exists \quad m_i \geq 0 \quad \text{with} \quad F_i = -m_i (e_i - e_{i-1}) \right\}, \tag{11}$$

The constraint $F_i = -m_i (e_i - e_{i-1})$ with $m_i \geq 0$ naturally depends on the specific energy $e$ used in the model. The later simply imposes the energy fluxes to be opposed to the energy gradient. We point out that the energy conservation is implicit in the flux formulation of the variational problem and doesn't need to be imposed as a constraint here.

## 3 Results

We have computed temperature, specific energy (energy per unit mass), and energy flux profiles for different prescribed atmospheric compositions from A. McClatchey et al. (1972) corresponding to tropical, mid-latitude summer, mid-latitude winter, sub-arctic summer, and sub-arctic winter conditions. We work with a fixed relative humidity profile (ratio of the partial pressure of water vapor to the equilibrium vapor pressure of water at a given temperature). Typical values of surface albedo are used: $\alpha = 0.1$ for tropical and mid-latitude conditions and $\alpha = 0.6$ for sub-arctic ones. The atmosphere is discretized in $N = 20$ vertical levels.

### 3.1 The effect of the constraint

We investigate the effect of the following energy terms on the constraint:

- Sensible heat $C_p T$;

- Geo-potential $gz$;

- Latent heat for a water vapor-saturated air $Lq_s(T)$, where $q_s$ is the mixing ratio at the saturation point. Since we work in pressure coordinates, it depends only on local temperature. This is a first attempt to take into account the effect of humidity without an explicit derivation of the humidity profile and water cycle. However, the radiative budget is still computed using a fixed standard relative humidity profile.

For illustration purpose, we can consider the case with only 2 layers. We note $F = m(e_1 - e_2)$ the net energy flux from layer 1 to layer 2, where $m$ is the mass mixing coefficient between layers (cf. figure 2). In this simple case, the entropy production writes $\sigma = F(1/T_2 - 1/T_1)$ and is limited by the constraint $m \geq 0 \iff F(e_1 - e_2) \geq 0$. We interpret the different energy terms on this constraint as follow:

- $e = C_p T$: $F \geq 0$ if $T_1 \geq T_2$. The constraint simply imposes the energy transport from hot to cold regions.

- $e = C_p T + gz$: $F \geq 0$ if $T_1 \geq T_2 + g(z_2 - z_1)/C_p$: the geopotential $gz$ limits the upward energy flux. We predict a warmer air at the bottom and a colder air at the top compared to the model with only sensible heat. Mechanically, we can

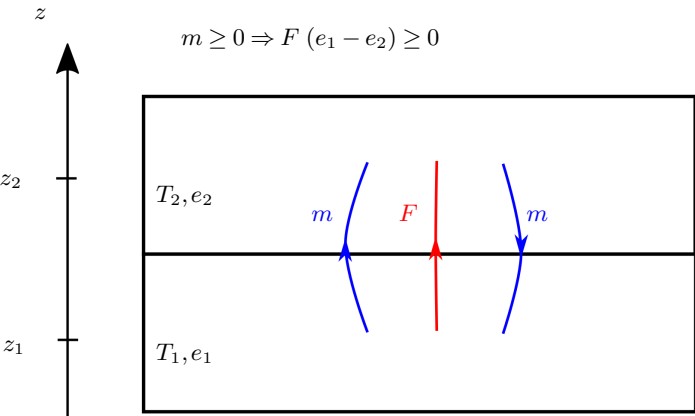

**Figure 2.** Energy exchanges between 2 layers of elevation $z_1$ and $z_2$ ($z_2 \geq z_1$), temperatures $T_1$ et $T_2$, and specific energy $e_1$ and $e_2$. We note $F = m(e_1 - e_2)$ the energy flux from the layer 1 to the layer 2, where $m$ is the mass mixing coefficient between the two layers.

see this as the expression of the fact that an air parcel from layer 1 may not have enough energy to move adiabatically to layer 2.

– $e = C_p T + gz + Lq_s$: $F \geq 0$ if $T_1 \geq T_2 + [g(z_2 - z_1) + L(q_s(T_2) - q_s(T_1))]/C_p$. Since $q_s(T)$ is an increasing function, $q_s(T_2) - q_s(T_1)$ has the same sign as $T_2 - T_1$. The temperature gradient is usually negative (i.e $T_2 \leq T_1$). Adding the latent heat at saturation makes the upward transport of energy less constrained. Consequently, the atmospheric temperature gradient weakens. Mechanically, we can see this as the fact that moist convection is easier than dry convection because of the transport of latent energy from the bottom to the top of the atmosphere.

## 3.2 General remarks

Various profiles are shown for tropical (figure 3) and sub-arctic winter (figure 4) conditions. The outputs of our constrained model are labelled by the energy terms taken into account in the constraint ($e = C_p T$, $e = C_p T + gz$ or $e = C_p T + gz + Lq_s$). The represented energy profiles are the energy corresponding to the constraint. For $C_p T$ we represent the profile $e = C_p T$, for $C_p T + gz$ we represent $e = C_p T + gz$, and for for $C_p T + gz + Lq_s$ we represent $e = C_p T + gz + Lq_s$. For $e = C_p T$, the specific energy is trivially more important for hot regions. For $e = C p_T + gz$, the geopotential adds energy to upper layers. For $e = C p_T + gz + Lq_s$, the latent heat term adds energy to more humid layers. The results of the "unconstrained" model of Herbert et al. (2013) are represented with the associated thermal energy $e = C_p T$ (rigorously speaking, the model is constrained by the global conservation of energy, but we will refer to it as "unconstrained" since no constraint is imposed on fluxes). Temperature profiles measured by A. McClatchey et al. (1972) are also represented for qualitative comparison and labelled as "reference".

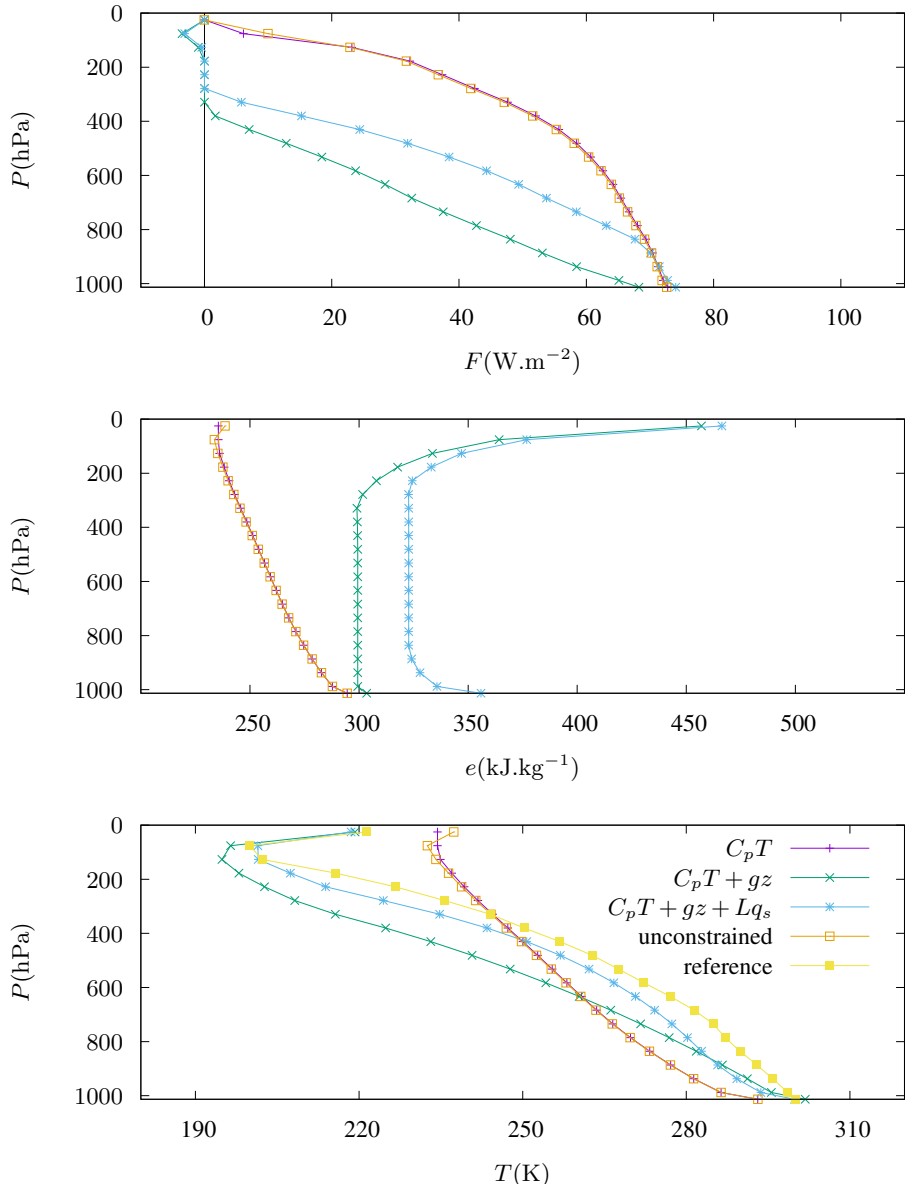

**Figure 3.** Energy flux $F$, specific energy $e$, and temperature $T$ for tropical atmospheric composition measured by A. McClatchey et al. (1972) and different expressions for energy on the constraint ($e = C_p T$, $e = C_p T + gz$ and $e = C_p T + gz + Lq_s$). The elevation is given in pressure level $P$. Results for the unconstrained model of Herbert et al. (2013) are represented. We also give the temperature profile corresponding to the measurements of A. McClatchey et al. (1972), labelled as "reference", for qualitative comparison.

For the unconstrained model, the energy flux is positive (i.e. upward) for the tropical (figure 3), and sub arctic winter (figure 4) atmospheric compositions in all the column, despite the energy gradient inversion in the upper layers of atmosphere. Therefore,

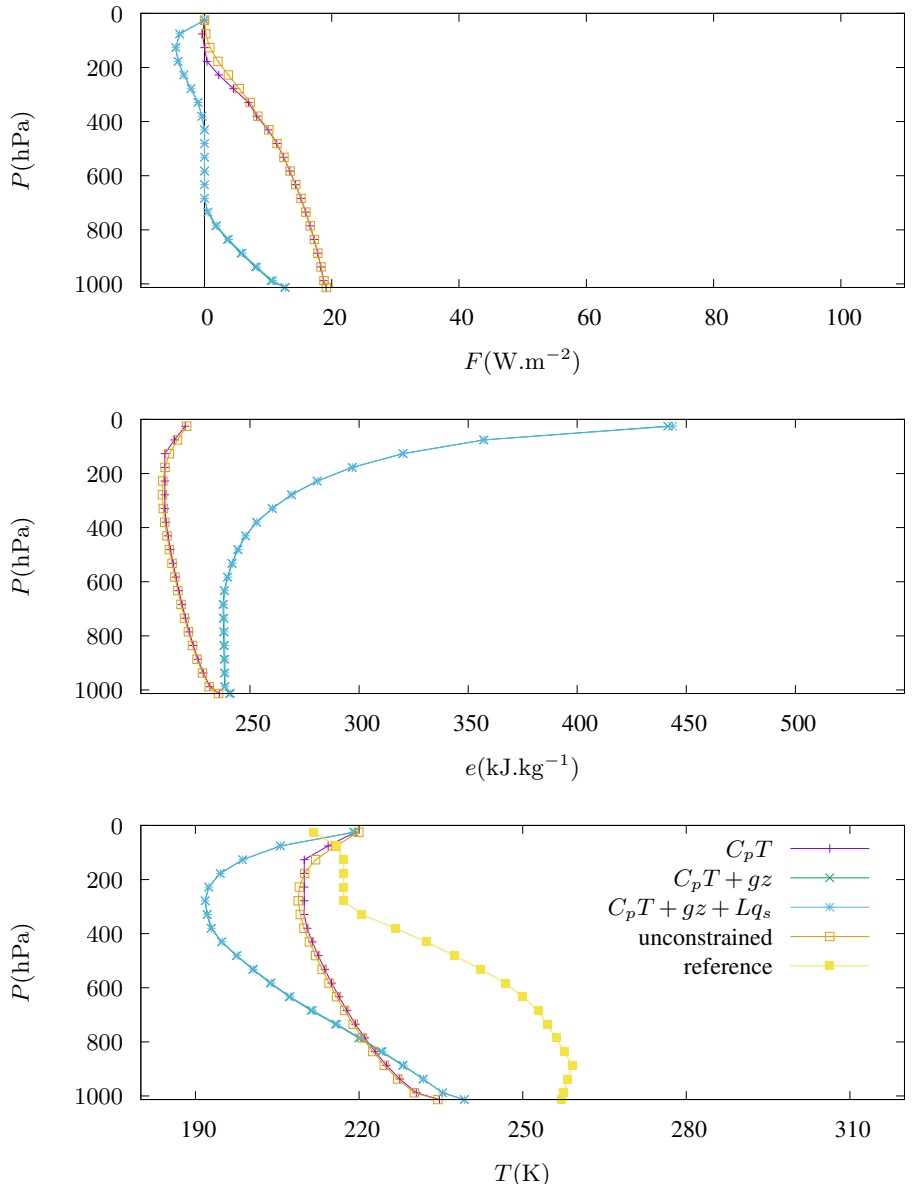

**Figure 4.** Energy flux $F$, specific energy $e$, and temperature $T$ for sub-arctic winter atmospheric composition measured by A. McClatchey et al. (1972) and different expressions for energy on the constraint ($e = C_p T$, $e = C_p T + gz$ and $e = C_p T + gz + Lq_s$). The elevation is given in pressure level $P$. Results for the unconstrained model of Herbert et al. (2013) are represented. We also give the temperature profile corresponding to the measurements of A. McClatchey et al. (1972), labelled as "reference", for qualitative comparison.

the flux is in the same direction as the energy gradient in this region. This also corresponds to local negative entropy production and is not physically relevant. Consequently, the upward flux is overestimated, and the temperature gradient is weak. As discussed above (part 2.1), the addition of the constraint $m_i \geq 0$ imposes the energy flux to be opposed to the specific energy gradient everywhere. As a consequence, the local entropy production is also constrained to be positive as in Ozawa and Ohmura (1997). But with an explicit account for the mass transport as explained above, we can account not only for sensible heat, but more generally for moist static energy transfers in a convective column. If an energy flux in this direction is not favourable in term of entropy production, it vanishes and we have stratification. When the geopotential term is considered (i.e. $e = C_p T + gz$ or $e = C_p T + gz + Lq_s$), we observe:

- An energy profile divided in 3 regions:

    1. An unstable surface layer with a decreasing energy profile;

    2. A neutral (slightly stable) mixed layer in the middle atmosphere with a vanishing energy gradient;

    3. An inversion layer at the top of the atmosphere where the energy is increasing with altitude.

- A vanishing energy flux (stratification) in the upper part of the atmosphere (around $P \simeq 300$ hPa for tropics (figure 3) and $P \simeq 700$ hPa for sub-arctic winter (figure 4)).

We note the thermal gradient is divided roughly by a factor of two when the geopotential is considered, which gives a more realistic temperature profile.

### 3.3   Comparison between profiles

#### 3.3.1   Model outputs for different climatic conditions

The constrained model is obviously sensitive to the water content. Considering $e = C_p T + gz$ or $e = C_p T + gz + Lq_s$ in the constraint gives approximately the same results for sub-arctic winter conditions (figure 4) (since $q_s(T)$ is weak for low temperatures) while predictions differ for tropical conditions (figure 3) (where $T$ and then $q_s(T)$ are more important). We have verified that the influence of the surface albedo explains the large part of temperature's modification when we compare different climatic conditions.

#### 3.3.2   Model output vs reference profile

Before discussing the differences between the outputs and observations, we point out that this conceptual model does not take into account some important physical processes:

- Insolation is assumed to be constant, fixed at $1368/4$ W.m$^{-2}$ for all conditions. But in reality, it varies with respect to seasons, diurnal cycle and latitude due to the Earth's geometry and obliquity. So, the model doesn't take into account the variation of the radiative budget because of these geometrical factors;

- Horizontal energy fluxes are not considered in this 1D description;

- The effect of clouds, that plays an important role in the absorption and emission of radiations (Dufresne and Bony, 2008), is not implemented in the radiative code.

Therefore, the aim of this study is not to give realistic values of temperature profiles nor vertical energy fluxes, but to give a
qualitative evaluation of the model. However, we can make some remarks on the comparison of our results to the reference temperature profiles. We observe that our model with $e = C_p T + gz + Lq_s$ provides better results for tropical conditions (figure 3) whereas the computed profiles are not so good for sub-arctic winter conditions (figure 4). Considering the previous remarks, one can explain the gap between our model and observations as follow. Constant insolation at the seasonal time scale is valid for tropics, but it varies strongly for high latitudes. So, our model is not adapted to represent a specific season at a high latitude
like sub-arctic winter conditions.

Tropical regions are submitted to strong vertical motion due to radiative heating. So horizontal energy fluxes are less important and the 1D vertical description may be more adapted to this case. In contrast, radiative heating is less important for the Arctic (especially in winter), so convection is weaker. Then, the representation of horizontal energy fluxes is essential at high latitudes since they play a major role in heat transport from hot equatorial regions to cold poles. This probably explains why
we underestimate temperature for high latitudes (figure 4).

### 3.4    Sensitivity to atmospheric composition

#### 3.4.1    Ozone

When the influence of $O_3$ is not considered on the radiative budget, we observe small changes for specific energy and large changes for temperature in the stratosphere (cf figure 5). Indeed, $O_3$ absorbs solar radiation at the top of the atmosphere
which induces heating of this region. It follows that the temperature in the high atmosphere is more important with ozone, and we even observe an inversion of the temperature gradient. It follows that less solar radiation heats the ground, resulting in smaller surface temperature. For our constrained model with $e = C_p T + gz + Lq_s$ including ozone, we observe a downward convective energy flux at the top of the atmosphere (figure 5). Ozone is therefore associated with heating from the top with an inversion of the temperature gradient and downward energy fluxes in the high atmosphere. This last effect only appears when
both geo-potential energy and $O_3$ are taken into account. When geo-potential is not considered, the upward energy flux is so overestimated that the effect is undetectable (figures 3, 4).

#### 3.4.2    Carbon dioxide

We also have performed the classic experiment of doubling $CO_2$ concentration (Randall et al., 2007). The climate sensitivity, defined as the surface temperature differences between computations with $[co2] = 560$ ppm and $[co2] = 280$ ppm, is reported
for the different atmospheric compositions in table 1. Conventional models usually represent various processes like water vapor, ice-albedo, lapse rate, and clouds feedbacks. They play an important role in amplifying the climate sensitivity (Forster

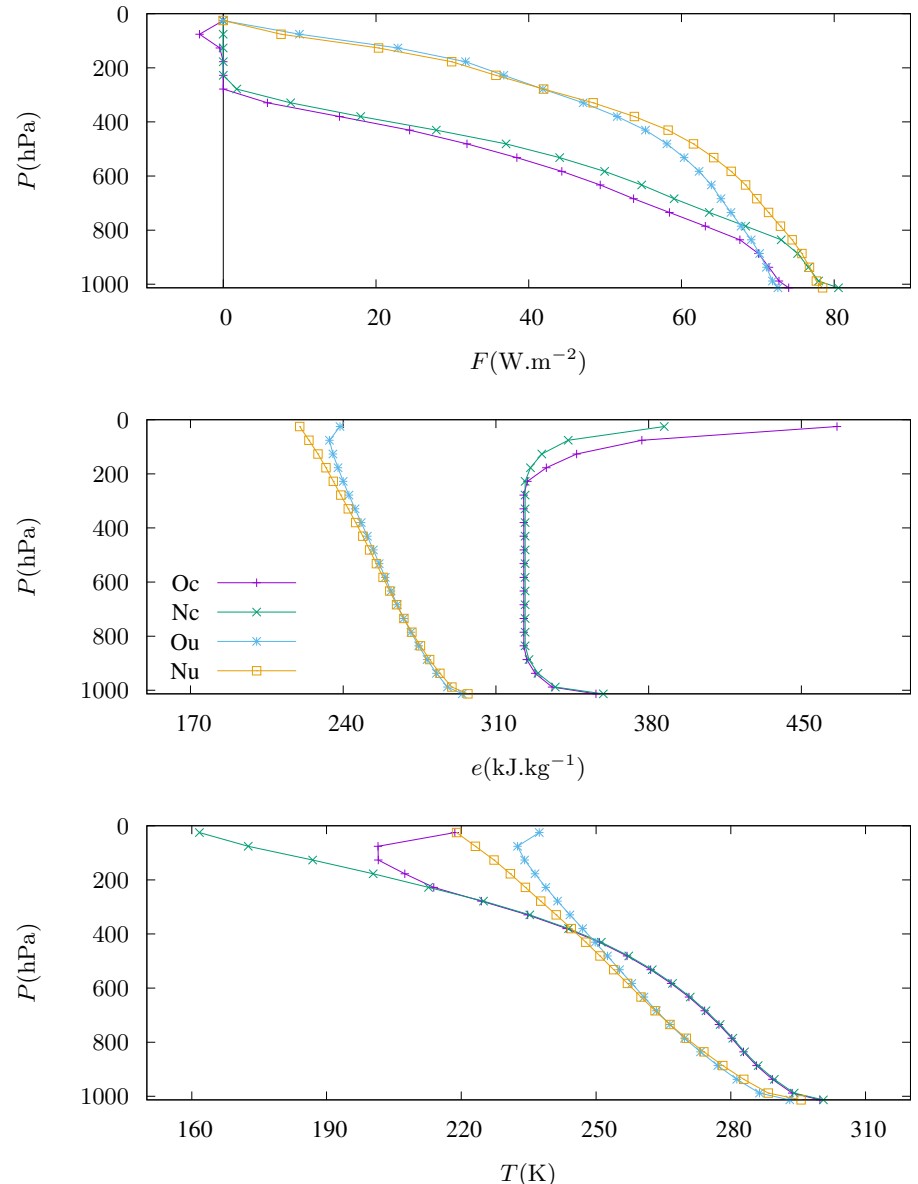

**Figure 5.** Energy flux $F$, specific energy $e$, and temperature $T$, with and without ozone, for the constrained model with $e = C_p T + gz + L q_s$ and for the unconstrained model and for tropical atmospheric composition of A. McClatchey et al. (1972). Oc : with $O_3$, constrained. Nc : without $O_3$, constrained. Ou : with $O_3$, unconstrained. Nu : without $O_3$, unconstrained.

and Gregory, 2006). When comparing our values with the literature, we must keep in mind that our model does not represent all those feedbacks. The lapse rate feedback is taken into account. Water vapor feedback is partially represented in a crude way by fixing relative humidity (changes in temperature have an impact on water content and change the radiative budget), but there is no explicit representation of the hydrological cycle. It is technically possible to include the ice-albedo feedback in a MEPM (Herbert et al., 2011a), but this is not the case here. Clouds are not represented in the model. So we will focus on the comparison between the constrained and unconstrained model using the same radiative scheme. Nevertheless, typical values of climate sensitivity for multi models averages with only relevant feedbacks are given (Dufresne and Bony, 2008) for qualitative comparison. For fixed absolute water profile, our values are compared to Dufresne and Bony (2008) accounting only for the lapse rate feedback, and for the fixed relative humidity, we compare them to Dufresne and Bony (2008) accounting with both lapse rate and water vapor feedbacks.

The sensitivity values computed here for the unconstrained model differ from Herbert et al. (2013). We have checked that it is only due to the fact that we use $N = 20$ atmospheric layers here instead of $N = 9$ in Herbert et al. (2013) (see annex B for the convergence of the algorithm with $N$). The climate sensitivity is higher for the constrained model than the unconstrained one (despite one exception with fix absolute moisture for Mid-latitude winter). This result may be interpreted as follow. If we start with a radiative forcing induced by a $CO_2$ doubling, it is the same for the two models since we fix identical atmospheric compositions and surface albedos. However, upward energy fluxes are limited for the constrained case which induces more important warming of the lower part of the atmosphere. The induced ground temperature elevation is, therefore, more important for the constrained model. This effect is observable when we look at the perturbation of energy flux, specific energy, and temperature (figure 6). The constrained model provides more realistic values of sensitivity for fixed relative humidity, particularly for tropics. Indeed, the sensitivity $1.60$ K computed in this case is closer to the literature values $2.1 \pm 0.2$ K (Dufresne and Bony, 2008) than the unconstrained model.

## 4  Discussion

MEPMs are different from the usual GCMs or EMICs. Generally, atmospheric models are based on:

1. Kinematics: equations describing how the fluid moves.

2. Dynamics: equations describing why the fluid moves. They are based on Navier-Stokes equations linking the fluid acceleration to the forces.

3. Thermodynamics: energy budget equation involving dissipation, radiations, phases changes, ...

4. An equation of state like perfect gas relation, and approximations like hydrostatic to simplify the problem.

5. Closure hypothesis and/or parametrizations to represent sub-grid processes.

In usual climate models, energy transport is then obtained after the computation of the velocity and other fields whereas in MEPMs, energy fluxes are computed implicitly without consideration on the dynamics. According to Dewar (2003) and Dewar

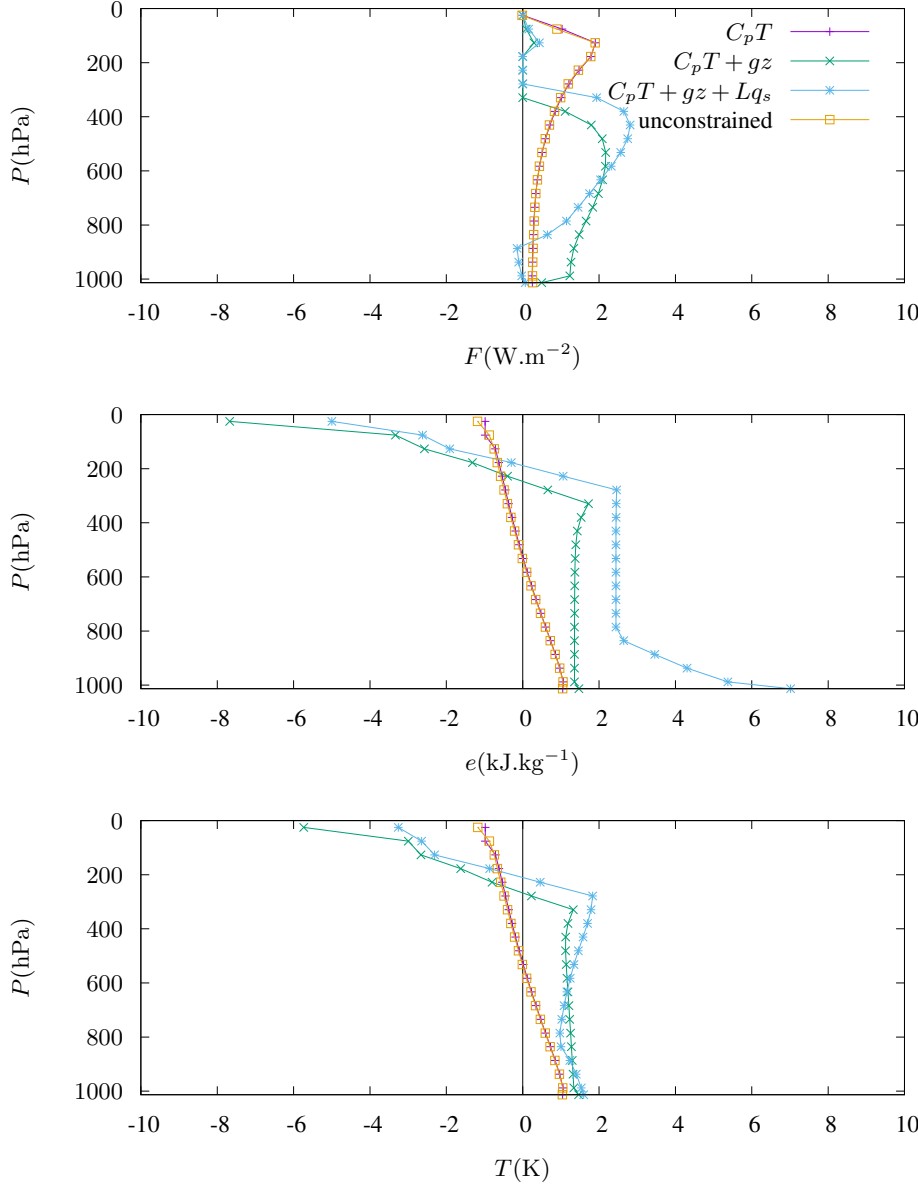

**Figure 6.** Differences of convective energy flux $F$, specific energy $e$, and temperature $T$ between $[co2] = 560$ ppm and $[co2] = 280$ ppm for tropical atmospheric composition of A. McClatchey et al. (1972) are represented for constrained model with $e = C_p T$, $e = C_p T + gz$ and $e = C_p T + gz + Lq_s$, and for the unconstrained model of Herbert et al. (2013).

**Table 1.** Climate sensitivity (warming (in K) of the surface due to a doubling of $CO_2$ concentration) of the constrained model with $e = C_p T + gz + Lq_s$, unconstrained model of Herbert et al. (2013), and literature (Dufresne and Bony, 2008). We give the values for different atmospheric compositions, and for fixed absolute or relative water vapor profiles.

| Conditions | | | Climate sensitivity | | |
|---|---|---|---|---|---|
| Moisture | Atmospheric composition (A. McClatchey et al., 1972) | Surface albedo | Unconstrained model (K) | Constrained model (K) | Literature (K) (Dufresne and Bony, 2008) |
| Absolute | Tropical | 0.1 | 0.90 | 1.06 | |
| | Mid-latitude summer | 0.1 | 0.79 | 0.93 | |
| | Mid-latitude winter | 0.1 | 0.46 | 0.24 | $0.4 \pm 0.3$ |
| | Sub-arctic summer | 0.6 | 0.53 | 1.43 | (lapse rate only) |
| | Sub-arctic winter | 0.6 | 0.20 | 0.31 | |
| Relative | Tropical | 0.1 | 1.04 | 1.60 | |
| | Mid-latitude summer | 0.1 | 0.97 | 1.30 | |
| | Mid-latitude winter | 0.1 | 0.82 | 1.19 | $2.1 \pm 0.2$ |
| | Sub-arctic summer | 0.6 | 0.15 | 0.39 | (lapse rate + water vapor) |
| | Sub-arctic winter | 0.6 | 0.09 | 0.19 | |

(2009), MEP might be viewed as a statistical inference (such as the information theory interpretation of Statistical Physics of Jaynes (1957)), rather than a physical law. From this point of view, MEP allows drawing predictions (heat fluxes, specific energy, and temperature profiles) from the partial knowledge of the radiative budget and the energy content, while the effect of the sub-grid scale processes is unknown. There are recent attempts to link MEP to other variational principles for dynamical systems/non-equilibrium statistical physics like the maximum of Kolmogorov-Sinaï entropy (Mihelich, 2015). Similar methods like the maximization of dynamical entropy or "Maximum Caliber" (Monthus, 2011; Dixit et al., 2018) may also be relevant to understand where MEP arises from a more formal point of view.

The gap between the model and observations is easily understood and explains why our 1D description with constant insolation is more adapted for tropics than arctic conditions. Further improvements are needed to solve these problems. Firstly, a more general 2 or 3-dimensional mass scheme transport is required. Secondly, it is formally possible to compute winds (Karkar and Paillard, 2015), moisture profiles, humidity fluxes using MEP. Finally, we need to include a time dependence in our model for seasonal or diurnal cycles. A long-term objective might be to construct an SCM, with a limited number of adjustable parameters.

## 5 Conclusions

We have investigated the possibility of computing the vertical energy fluxes and temperature in the atmosphere using the MEP closure hypothesis into a simple climate model. The fluxes are then computed in an implicit way, which avoids tuning

parameters. Contrary to (Ozawa and Ohmura, 1997; Pujol and Fort, 2002; Herbert et al., 2013; Pascale et al., 2012), we have given a description of how the energy is transported. This paper provides the first attempt, up to the authors' knowledge, to introduce such a representation in a MEPM. Different energy terms can be considered: sensible heat, geo-potential and latent heat for a saturated profile. We have shown that this better energetic description of convection allows obtaining more physically

relevant temperature, specific energy, and energy fluxes profiles, still without any adjustable parameter for the dynamics. In particular, considering geo-potential leads to stratification in the upper atmosphere and allows us to reproduce a temperature gradient closer to the observed one. We have investigated the sensitivity of the model when the atmosphere's composition is modified. The results were compared to previous MEPM and literature. Our model is more sensitive to $CO_2$ than in Herbert et al. (2013) because the geo-potential limits upward energy fluxes.

We hope that the present model may be helpful to construct SCMs with a reduced number of adjustable parameters.

*Code availability.* A Python code, based on the module scipy.optimize, that reproduce the results presented in this paper can be found here https://doi.org/10.5281/zenodo.2597543.

## Appendix A:  Computing geo-potential

We show here how to compute the geopotential and the dry static energy $e_i^d = C_p T_i + g z_i$ in term of temperatures. One first

writes

$$z_i = z_i - z_{i-\frac{1}{2}} + \sum_{j=1}^{i-1} \Delta z_j \tag{A1}$$

where $\Delta z_j = z_{j+\frac{1}{2}} - z_{j-\frac{1}{2}}$ is the height of the layer $j$. So, if $\rho$ is the density of the air, $R$ is the specific air constant and we assume the atmosphere is an ideal gas at hydrostatic equilibrium

$$g\Delta z_j = \int_{z_{j-\frac{1}{2}}}^{z_{j+\frac{1}{2}}} g\,\mathrm{d}z = -\int_{p_{j-\frac{1}{2}}}^{p_{j+\frac{1}{2}}} \frac{\mathrm{d}p}{\rho} = -\int_{p_{j-\frac{1}{2}}}^{p_{j+\frac{1}{2}}} RT\,\frac{\mathrm{d}p}{p}. \tag{A2}$$

Then, we can compute the mean elevation of a layer with two possible prescriptions:

– Isothermal layers ($T = T_j$ in the integrand) :

$$g\Delta z_j = RT_j \ln\left(\frac{p_{j-\frac{1}{2}}}{p_{j+\frac{1}{2}}}\right). \tag{A3}$$

So the geopotential reads

$$g z_i = R\left[T_i \ln\left(\frac{p_{i-\frac{1}{2}}}{p_i}\right) + \sum_{j=1}^{i-1} T_j \ln\left(\frac{p_{j-\frac{1}{2}}}{p_{j+\frac{1}{2}}}\right)\right]. \tag{A4}$$

- Dry isentropic layers ($T = T_j \left( \frac{p}{p_j} \right)^{\frac{R}{C_p}}$ in the integrand):

$$g\Delta z_j = C_p T_j \left[ \left( \frac{p_{j-\frac{1}{2}}}{p_j} \right)^{\frac{R}{C_p}} - \left( \frac{p_{j+\frac{1}{2}}}{p_j} \right)^{\frac{R}{C_p}} \right]. \tag{A5}$$

So the geo-potential reads

$$gz_i = C_p \left[ T_i \left( \left( \frac{p_{i-\frac{1}{2}}}{p_i} \right)^{\frac{R}{C_p}} - 1 \right) + \sum_{j=1}^{i-1} T_j \left( \left( \frac{p_{j-\frac{1}{2}}}{p_j} \right)^{\frac{R}{C_p}} - \left( \frac{p_{j+\frac{1}{2}}}{p_j} \right)^{\frac{R}{C_p}} \right) \right]. \tag{A6}$$

In both cases, for imposed pressure levels, we obtain the following expression of the specific energy

$$e_i^d \equiv \sum_{j=0}^{N} (C_p \delta_{ij} + G_{ij}) T_j \equiv \sum_{j=0}^{N} E_{ij}^d T_j, \tag{A7}$$

where $\delta_{ij}$ is the Kronecker symbol, $G$ and $E^d$ are constant matrices.

## Appendix B: Resolution

In order to solve the optimization problem (11), we express it in Lagrangian formalism, assuming strong duality holds (Boyd

and Vandenberghe, 2004). We therefore search the critical points of the Lagrangian associated to this problem

$$\mathcal{L} = \sigma - \sum_{i=1}^{N} \mu_i\, m_i \qquad \text{with} \qquad \begin{cases} m_i \geq 0 \\ \mu_i \geq 0 \end{cases} \qquad \text{and} \qquad \mu_i\, m_i = 0 \qquad i = 1, ..., N. \tag{B1}$$

Where $\mu_1, ..., \mu_N$ are Lagrange multipliers associated to the constraint (mass flux positivity). In order to formulate the problem in term of energy fluxes $F$, we must express the inverse temperature $X = 1/T$ and the mass mixing $m$ with $F$.

We use an iterative method to solve this nonlinear optimization problem:

1. We linearize the radiative budget and specific energy around a given temperature profile.

2. The entropy production and constraints are then quadratic forms of energy fluxes that can be solved numerically.

3. We reiterate step 1 by linearising around the temperature profile obtained in step 2 until convergence.

This is a rather standard procedure for optimization though there is no guarantee of finding the global solution in case

of multiple local maxima. To overcome this issue, we start with various random initial temperature profiles that may lead to different local maxima. In the end, we retain only the best maximum which is assumed to be the maximum of entropy production.

## B1 Flux-temperature relation

The energy balance equation in stationary state can be written as follow

$$\mathcal{R}_i(X) + F_i - F_{i+1} = 0. \tag{B2}$$

At each iteration, we linearise the radiative budget $\mathcal{R}_i(X)$ around a reference temperature profile $X^0$ :

$$5 \quad \mathcal{R}_i(X) \simeq \mathcal{R}_i(X^0) + \sum_{j=0}^{N} R_{ij} \left( X_j - X_j^0 \right), \tag{B3}$$

where $R$ is a square matrix of size $N$ and $\mathcal{R}(X^0)$ is the radiative budget for the profile $X^0$. If we assume $R$ to be invertible, the energy flux can be computed with

$$X_i = X_i^0 - \sum_{j=0}^{N} R_{ij}^{-1} \left( F_j - F_{j+1} + \mathcal{R}_j(X^0) \right). \tag{B4}$$

## B2 Flux-mass relation

10 We first consider the dry static energy $e^d = C_p T + gz$. Considering the atmosphere is an ideal gas at hydrostatic equilibrium, and for prescribed pressure levels, layer volume depends only on temperature. Therefore, the elevation of a layer is a function of temperatures of layers below only and we can express the energy of a layer as (see Annexe A)

$$e_i^d = \sum_{k=0}^{N} E_{ik}^d \, T_k, \tag{B5}$$

where $E^d$ is a square, triangular matrix of size $N$. If we linearise around $X^0$, one obtains

$$e_i^d \simeq \sum_{k=0}^{N} \frac{E_{ik}^d}{X_k^{0\,2}} (2X_k^0 - X_k). \tag{B6}$$

Using equation (B4), it gives the expression of energy in term of flux

$$e_i^d(F) \simeq \sum_{k=0}^{N} \frac{E_{ik}^d}{X_k^{0\,2}} \left( X_k^0 + \sum_{j=0}^{N} R_{kj}^{-1} \left( F_j - F_{j+1} + \mathcal{R}_j(X^0) \right) \right). \tag{B7}$$

We also can take into account the latent heat for a water vapor-saturated atmosphere. The mixing ratio at the saturation point

20 $q_s$ depends only of temperature $T$ (in K) and pressure $p$ (in Pa). It is given by the Bolton equation (Bolton, 1980):

$$q_s(T,p) = \frac{622.0 h_s(T)}{p - h_s(T)} \qquad \text{with} \qquad h_s(T) = 6.112 \exp\left( \frac{17.62(T - 273.15)}{T - 30.03} \right). \tag{B8}$$

where $h_s$ is the mixture's saturation vapor pressure (in Pa). At fixed pressure, the moist static energy at saturation $e^s = C_p T + gz + Lq_s$ of layers is only function of temperatures. If we linearise around the profile $X^0$,

$$e_i^s = e_i^d + Lq_s\left( \frac{1}{X_i} \right) \simeq e_i^d + Lq_s\left( \frac{1}{X_i^0} \right) - L \left. \frac{\partial q_s}{\partial T} \right|_{\frac{1}{X_i^0}} \frac{X_i - X_i^0}{X_i^{0\,2}}, \tag{B9}$$

where $\delta_{ik}$ is the Kronecker symbol. Then, we can use the same reasoning as for the dry static heat and replace the matrix $E^d$ by $E^s$ to consider the effect of latent energy for a saturated moisture profile. However, the radiative budget is still computed with reference water vapor profiles. In the following, $e$ can represent $e^d$ or $e^s$

## B3 Constraint

5 By multiplying both side of

$$F_i = -m_i \left( e_i - e_{i-1} \right) \tag{B10}$$

by $(e_i - e_{i-1})$, we obtain

$$F_i \left( e_i - e_{i-1} \right) = -m_i \left( e_i - e_{i-1} \right)^2 \tag{B11}$$

So the constraint $m_i \geq 0$ is equivalent to

10 $\quad \alpha_i(F) \equiv -F_i \left( e_i(F) - e_{i-1}(F) \right) \geq 0,$ $\tag{B12}$

## B4 Associated Lagrangian in flux space

Using the linearised energy budget (B4) and the constraint (B12), the problem (11) is supposed to be equivalent to the search of critical points of the following Lagrangian

$$\mathcal{L}(F,\mu) = \sigma(F) - \sum_{i=1}^{N} \mu_i \, \alpha_i(F)$$

$$= \sum_{i=0}^{N} X_i \, \left( F_i - F_{i+1} \right) - \sum_{i=1}^{N} \mu_i \, \alpha_i \tag{B13}$$

$$\simeq \sum_{i=0}^{N} \left( X_i^0 - \sum_{j=0}^{N} R_{ij}^{-1} \left( F_j - F_{j+1} + \mathcal{R}_j(X^0) \right) \right) \left( F_i - F_{i+1} \right) - \sum_{i=1}^{N} \mu_i \, \alpha_i(F)$$

$$\tag{B14}$$

while respecting the Karush-Kuhn-Tucker (KKT) conditions

$$\frac{\partial \mathcal{L}}{\partial F_i} = 0 \qquad \text{with} \qquad \begin{cases} \alpha_i(F) \geq 0 \\ \mu_i \geq 0 \end{cases} \qquad \text{and} \qquad \mu_i \, \alpha_i(F) = 0 \qquad i = 1,...,N. \tag{B15}$$

20 The problem is solved numerically by using an Interior point method Boyd and Vandenberghe (2004).

## B5 Convergence of the algorithm

In practice, the algorithm may fail to find the global optimum for large $N$ ($\geq 50$) but is robust for $N \leq 40$. As $N$ increases, the algorithm converges rapidly to a solution (figure B1). The choice $N = 20$ is a good compromise between computation time and resolution.

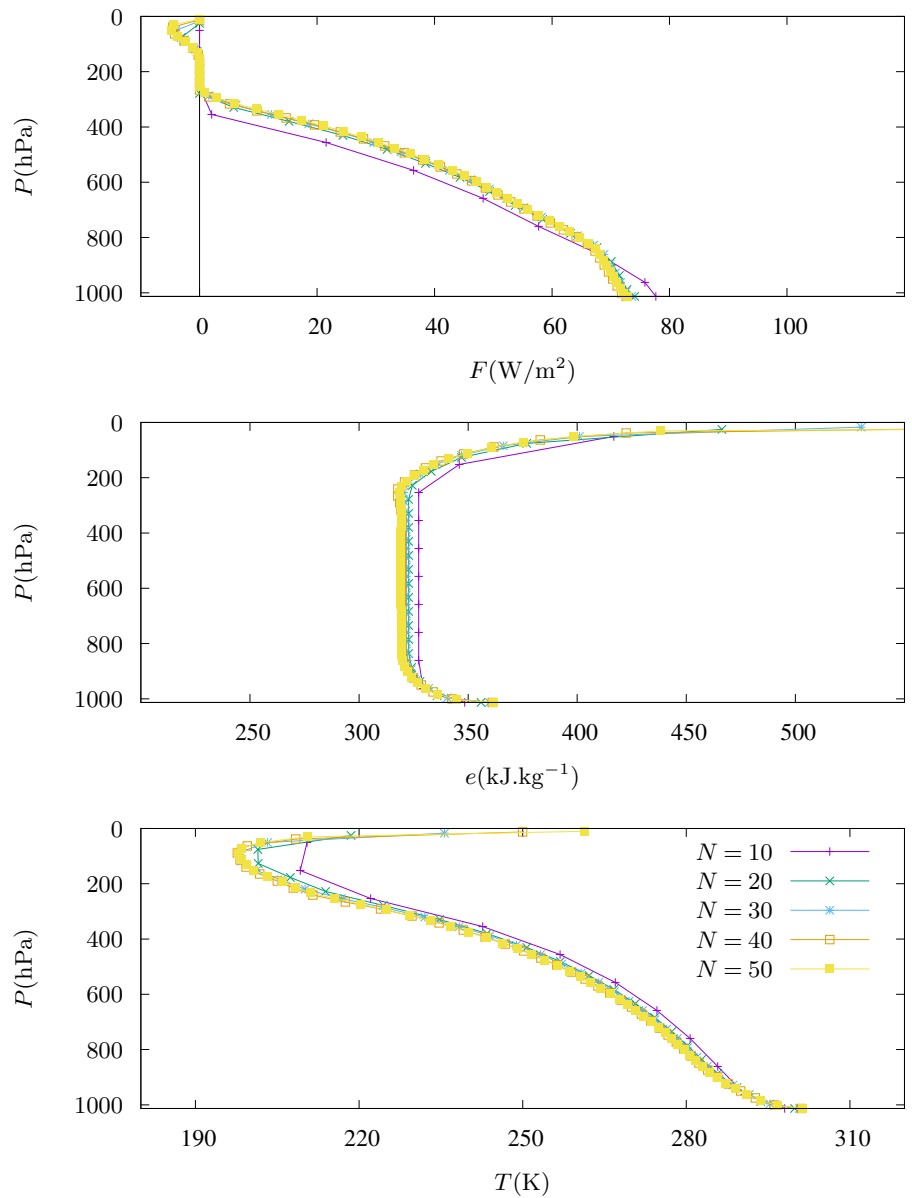

**Figure B1.** Energy flux $F$, specific energy $e$, and temperature $T$ computed by our constrained model with $e = C_p T + gz + Lq_s$ for imposed tropical atmospheric composition measured by A. McClatchey et al. (1972), and various resolution $N$.

*Competing interests.* Authors declare no conflict of interest

*Acknowledgements.* We thank Stan Schymanski and the anonymous reviewer for their useful comments that helped us to improve the manuscript.

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
