# Peer review of "A Radiative-Convective Model based on constrained Maximum Entropy Production"

_Earth System Dynamics, 2018_

## Referee Comment (RC2)

Stan Schymanski (stan.schymanski@env.ethz.ch)

December 23, 2018

I have read the paper with great interest and tried to follow and verify the authors' line of arguments. Below, I try to give an untainted and unbiased view of the manuscript, but of course it represents my personal perspective, subject to my own limitations.

**1 Summary of key results**

The manuscript entitled "A Radiative Convective Model based on constrained Maximum Entropy Production" by Labarre, Paillard and Dubrulle follows on previous articles investigating the usefulness of the Maximum Entropy Production (MEP) principle for climate modelling [e.g. Herbert et al., 2011a,b, 2013, Herbert and Paillard, 2014], with two of the co-authors overlapping between articles. As in previous papers, the authors use the MEP principle as a closure scheme to compute energy fluxes between vertical air layers in a simple climate model (SCM). If I understood correctly, they add consideration of a gravitational field and latent heat flux to the approach by Herbert et al. [2013] and show that the resulting vertical temperature profiles are closer to observations compared with the previous approach for tropical conditions (e.g. Fig. 3c), but not for sub-arctic conditions (e.g. Fig. 4c). The authors also investigate the effect of ozone and different $CO2$ concentrations on the resulting vertical profiles of temperature and energy fluxes, but I was neither able to verify how these effects were implemented nor assess in how far they resemble observed effects. Table 1 suggests that the climate sensitivities to $CO2$ in all simulations was quite different to those presented in another paper, based on an Earth System Model. The authors explain some of the discrepancies between simulation results and observations by various simplifying assumptions in their model.

**2 General assessement and recommendation**

The linguistic shortcomings, already pointed out by Referee 1, were a bit distracting and so plentiful that I gave up highlighting them, in order to focus more

on the content. If this manuscript was going to be accepted for publication, it would need a serious revision for language and typography.

The promise of the MEP principle to reduce the need for empirical parametrization and/or model calibration is huge and hence I generally welcome attempts to build MEP-based models and evaluate them rigorously. However, the work presented here left me puzzled in many places about its thermodynamic basis.

As in previous work, the authors formulate entropy production as an energy flux between two layers divided by the temperature of one of the layers. However, here, the energy flux comprises not only sensible heat, but also water vapour and potential energy. One of the basic textbook expressions of entropy production is that it relates to the rate by which a given thermodynamic potential gradient is depleted, e.g. temperature, pressure, or chemical potential, in some treatments also gravitational potential [e.g. Kondepudi and Prigogine, 1998, Eq. 10.1.9]. This is expressed as a product of a thermodynamic force and a thermodynamic flow, e.g. by a reaction rate multiplied by the affinity of the reaction divided by temperature or a heat flux divided by the temperature difference of two systems. In the present paper, mass flux is multiplied by the sum of thermal, gravitational and latent energy (Eqs. 1, 2 and 5), but I cannot draw the connection of these equations to the relevant thermodynamic formulations. For example, I do not see how the usual $TS$ term relates to $c_p T_i$, as $S \neq c_p$, or how $L q_i$ is related to the classical $\mu N$. I am also missing the $PV$ term in the formulation of total energy, and the consideration of volume work. Clearly, the authors should refer to established literature to help the reader reconcile their equations with thermodynamic theory.

I am also very confused about the combination of energy balance (Eq. 3) and the radiative budget (Eq. 4), as it suggests that all shortwave radiation reaching a given layer is absorbed and converted to either longwave radiation or one of the other energy terms. I suppose that $R_i$ should be the *absorbed* shortwave radiation, but then the equation still misses the absorbed longwave radiation, which is the main reason for the greenhouse effect. The paper does not specify at all how $SW_i$ and $LW_i$ are calculated.

I find even more puzzling the assumption that $m_i \geq 0$ (P6L1), termed as the "mechanical constraint". On P4L4, the authors define $m_i$ as either "the upward mass flux leaving the layer $i - 1$" or "the downward mass flux coming to the layer $i - 1$". This does not make any sense to me, as according to this wording, both would refer to the flux across the upper boundary of Layer $i - 1$ but would have opposite signs. Whatever the correct definition of $m_i$, I do not see how setting it to a positive value represents a mechanical constraint and how this is reconcilable with the statement on P11L14 about a "downward convective energy flux" in the case of light absorption by ozone at the top of the atmosphere.

Last but not least, the article lacks a data availability statement. See www.earth-system-dynamics.net/about/data_policy.html. Without the code and data used to generate the results and given the above-mentioned inclarities I am afraid that the manuscript is of very limited use.

**2.1 Recommendation**

While my above comments seem rather negative, I still believe that the paper could be a valuable contribution to the scientific literature, if the authors drew clear connections between their formulations and classical thermodynamic formulations, explaining all simplifying assumptions and if they provided all the information needed to reproduce their results. More rigorous comparison of simulations with observations would also be helpful. Since this, along with the necessary clean-up for language and typography, would require considerable effort, I would recommend a major revision or a recommendation for re-submission.

**3 Specific comments**

- Eq. 4: What about incoming longwave, reflection and re-absorption? Why does SW depend on T? What about LW emissivity?

- P5L8: What does 'non local dependence' mean?

- P5L21: This does not make sense, as mass flux can go in both directions.

- P6L14: What does 'absolute ratio moisture at saturation' mean? What are the units of $q_s$?

- P6L15: How is pressure prescribed? How does it vary with height?

- P7L10: What is the thermodynamic force driving the flux? Please explain why flux can be in the opposite direction to the energy gradient.

- Fig. 3: Is this for the tropical latitudes? Please specify what the data in each figure corresponds to and how it was generated.

- P10L26: Why would insolation vary more at high latitudes? It would vary more at the seasonal scale, but less at the diurnal scale. Which variation is more important for the profiles simulated here?

- P11L1-5: I do not understand the first sentence at all and the whole paragraph appears rather hand-waving to me. Why does the current version give worse results for high latitudes than Herbert et al. [2013]? More analysis and explanation would be necessary to generate any definitive insights here.

- Table 1: Where the literature results based on the same albedo values? How were these values chosen?

- P13L1-2: This sentence is not clear. Was it a discretization effect or not?

- P15L29: Observation-based temperature profiles were presented, but what are realistic 'energy content, and energy fluxes profiles'?

- References: Why are the references not sorted by author name?

**References**

C. Herbert and D. Paillard. Predictive Use of the Maximum Entropy Production Principle for Past and Present Climates. In R. C. Dewar, C. H. Lineweaver, R. K. Niven, and K. Regenauer-Lieb, editors, *Beyond the Second Law*, Understanding Complex Systems, pages 185–199. Springer Berlin Heidelberg, 2014. ISBN 978-3-642-40153-4 978-3-642-40154-1. doi: 10.1007/978-3-642-40154-1_9. URL http://link.springer.com/chapter/10.1007/978-3-642-40154-1_9.

C. Herbert, D. Paillard, and B. Dubrulle. Entropy production and multiple equilibria: the case of the ice-albedo feedback. *Earth System Dynamics*, 2 (1):13–23, Feb. 2011a. ISSN 2190-4987. doi: 10.5194/esd-2-13-2011. URL http://www.earth-syst-dynam.net/2/13/2011/.

C. Herbert, D. Paillard, M. Kageyama, and B. Dubrulle. Present and Last Glacial Maximum climates as states of maximum entropy production. *Quarterly Journal of the Royal Meteorological Society*, 137(657):1059–1069, Apr. 2011b. ISSN 0035-9009. doi: 10.1002/qj.832. URL http://rmets.onlinelibrary.wiley.com/doi/full/10.1002/qj.832.

C. Herbert, D. Paillard, and B. Dubrulle. Vertical Temperature Profiles at Maximum Entropy Production with a Net Exchange Radiative Formulation. *Journal of Climate*, 26(21):8545–8555, 2013. URL http://journals.ametsoc.org/doi/abs/10.1175/JCLI-D-13-00060.1.

D. Kondepudi and I. Prigogine. *Modern Thermodynamics, From Heat Engines to Dissipative Structures*. John Wiley & Sons, Chichester, 1998.

---

## Referee Comment (RC1) · Anonymous Referee #1 · 15 Nov 2018

**Overview**

The authors build on the model developed by Herbert et al (2013) (I note that the second and third authors have contributed to the earlier paper) to use the MEP conjecture to develop profiles for convective fluxes and temperatures. They add a further constraint, in explicitly expressing the energy transferred by convection as an upward and downward mass-flux times a quantity resembling moist static energy or some of its components. The atmospheric profiles that they obtain are compared with observations for various regions and sensitivities to atmospheric composition are obtained.

The paper is an interesting addition to work already published on MEP, and I would recommend publication subject to the comments below.

The paper suffers from poor presentation. It would benefit greatly from extensive copyediting by someone with a good level of scientific English, paying particular attention to words that have a different meaning in French and in English, and to the use of the definite article. (In some places I wonder if Google Translate deserves credit as a co-author!). There are also occasional inconsistent uses of the decimal comma. There are also rather odd spaces before commas in citations, parentheses for citations in a context where only the date of publication should be in parentheses and on occasion no spaces either side of a full stop separating sentences.

It is now unusual to see SI units separated by a full stop, but I presume that the journal has a house style which will state whether this is permitted or not. Likewise, atmospheric pressure is expressed here in terms of mB rather than the more usual hPa.

I only explicitly mention a handful of corrections to the language used below, where I particularly wish to bring a point to the authors' attention.

Minor remarks

L. 7 The word "ensemble" in climate science generally refers to a set of perturbed climate models which seek to establish the reliability of a forecast and is confusing here.

L. 16 Not just geometric – some models may use varying albedo.

L.19 It is not just the opacity of the surface, but the relative transparency of the atmosphere that is relevant.

L. 34 I would not regard "the absence of dynamics and/or the validity of MEP" as "one" reason to criticise MEP models.

Pages 8, 9, 12, 14

The captions to Figures 3, 4, 5 and 6 describe the plots in a different order to that

presented, which is mildly irritating.

**Page 9**

L. 13 – multiple energy profiles are presented; different constraints result in different characteristics. The authors should be more specific as to what they are describing.

L. 20. Clearly some thermal capacity is taken into account somwhere as C\_p is not zero!

L. 2 If the discretisation effect is important, have the authors satisfied themselves that N=20 is sufficient for their purposes?

**Page 16**

L. 14 I am concerned about the linearisation assumption. Radiation emitted by a layer will have a quartic dependence on temperature, so the linearisation will only be valid for a small perturbation. Is it possible to solve for T as a function of R other than by inverting a linear matrix?

---

## Author Comment (AC1) · 19 Jan 2019

Overview: The authors build on the model developed by Herbert et al (2013) (I note that the second and third authors have contributed to the earlier paper) to use the MEP conjecture to develop profiles for convective fluxes and temperatures. They add a further constraint, in explicitly expressing the energy transferred by convection as an upward and downward mass-flux times a quantity resembling moist static energy or some of its components. The atmospheric profiles that they obtain are compared with observations for various regions and sensitivities to atmospheric composition are obtained. The paper is an interesting addition to work already published on MEP, and I would recommend publication subject to the comments below.

Authors response 1:

We would like to thank the referee for reading our manuscript. We have read his remarks with great interest.

The paper suffers from poor presentation. It would benefit greatly from extensive copyediting by someone with a good level of scientific English, paying particular attention to words that have a different meaning in French and in English, and to the use of the definite article. (In some places I wonder if Google Translate deserves credit as a co-author!).

Authors response 2:

Both referees have noted numerous grammatical errors. We want to apologize for that. We will do our best to correct the manuscript.

There are also occasional inconsistent uses of the decimal comma. There are also rather odd spaces before commas in citations, parentheses for citations in a context where only the date of publication should be in parentheses and on occasion no spaces either side of a full stop separating sentences. It is now unusual to see SI units separated by a full stop, but I presume that the journal has a house style which will state whether this is permitted or not. Likewise, atmospheric pressure is expressed here in terms of mB rather than the more usual hPa.

Authors response 3:

We will also perform a thorough check of the units used in the manuscript and reread

the manuscript preparation guidelines to correct such errors.

I only explicitly mention a handful of corrections to the language used below, where I particularly wish to bring a point to the authors' attention. Minor remarks Page 2 L. 7 The word "ensemble" in climate science generally refers to a set of perturbed climate models which seek to establish the reliability of a forecast and is confusing here.

Authors response 4:

We will replace the word "ensemble" by "set" to avoid confusion.

L. 16 Not just geometric - some models may use varying albedo. L.19 It is not just the opacity of the surface, but the relative transparency of the atmosphere that is relevant.

Authors response 5:

We thank the referee for mentioning it, we will add these precisions in the next version of the manuscript.

L. 34 I would not regard "the absence of dynamics and/or the validity of MEP" as "one" reason to criticise MEP models.

СЗ

Authors response 6:

We agree with the reviewer. Still, this is often the main critic arising from the "Fluid dynamics" community, and it is difficult to ignore it altogether. We will write "have been criticized" instead of "can be criticized".

Pages 8, 9, 12, 14 The captions to Figures 3, 4, 5 and 6 describe the plots in a different order to that presented, which is mildly irritating.

Authors response 7:

We will present the captions of the figures in the same order as the figures themselves.

**Page 9**

*L.* 13 - multiple energy profiles are presented; different constraints result in different characteristics. The authors should be more specific as to what they are describing.

Authors response 8:

This concerns page7 L.13. The represented energy profiles are the energy corresponding to the constraint. For  $C_pT$  we represent the profile  $e = C_pT$ , for  $C_pT + gz$ we represent the profile  $e = C_pT + gz$ , and for for  $C_pT + gz + Lq_s$  we represent the profile  $e = C_pT + gz + Lq_s$ . For  $e = C_pT$ , the energy per unit mass is trivially more important for hot regions. For  $e = C_pT + gz$ , the geopotential adds energy to upper layers. For  $e = C_pT + gz + Lq_s$ , the latent energy term adds energy to more humid layers. In all cases, the constraint imposes the direction of the flux (opposed to the energy gradient). If an energy flux in this direction is not favorable in term of entropy production, it vanishes and we have stratification. We will add these precisions in the next version of the manuscript.

L. 20. Clearly some thermal capacity is taken into account somwhere as  $C_p$  is not zero!

Authors response 9:

This concerns page10 L.20. The reviewer is right. We are here confusing thermal inertia (the term with time variations of temperature  $C_p \frac{\partial T}{\partial t}$  which is equal to 0 in the stationary state) and thermal capacity. We will erase this point because it is not relevant to the comparison with reference profiles.

**Page 13**

*L*. 2 If the discretisation effect is important, have the authors satisfied themselves that *N*=20 is sufficient for their purposes?

Authors response 10:

We have been a little bit to fast on this point, but we have verified that N=20 is sufficient for our purposes. We will add the figure below that shows the temperature profiles for different values of N (for tropics) in the article. We observe that the method converges after  $N \simeq 15$ .

**Page 16**

L. 14 I am concerned about the linearisation assumption. Radiation emitted by a layer will have a quartic dependence on temperature, so the linearisation will only be valid

for a small perturbation. Is it possible to solve for T as a function of R other than by inverting a linear matrix?

Authors response 11:

We are not linearising the radiative budget. We are sorry for this misunderstanding. We are just using an iterative procedure to solve the full non-linear problem:

1) We linearize around a reference temperature profile.

2) Then we solve the linear constrained optimization problem.

3) Finally, we reiterate 1) by linearizing around the solution computed in 2) until convergence.

This is a rather standard procedure for optimization though there is no guarantee of finding the global optimum in case of multiple local maxima. In practise, the algorithm may fail for large N ( $N \simeq 50$ ) but is robust for  $N \leq 40$ . We will explain this point more clearly in the future version of the manuscript.

Fig. 1. Temperature vs dimensionless elevation for different numbers of layers N.

---

## Author Comment (AC3) · 6 Mar 2019

Comments to the Editor: Dear Editor,

We would like to ask some supplementary delay to complete some details in the manuscript and to furnish a clear code which reproduces some of the results. Is it possible to postpone the deadline to the 21st March?

Sincerely yours,

Vincent Labarre
* * *

---

## Author Comment (AC2)

**Responses to the comments of Stan Schymanski**

January 19, 2019

*I have read the paper with great interest and tried to follow and verify the authors' line of arguments. Below, I try to give an untainted and unbiased view of the manuscript, but of course it represents my personal perspective, subject to my own limitations.*

**Authors response 1:** We thank the reviewer for his interest for our manuscript.

**1) Summary of key results**

*The manuscript entitled "A Radiative Convective Model based on constrained Maximum Entropy Production" by Labarre, Paillard and Dubrulle follows on previous articles investigating the usefulness of the Maximum Entropy Production (MEP) principle for climate modelling [e.g. Herbert et al., 2011a,b, 2013, Herbert and Paillard, 2014], with two of the co-authors overlapping between articles. As in previous papers, the authors use the MEP principle as a closure scheme to compute energy fluxes between vertical air layers in a simple climate model (SCM). If I understood correctly, they add consideration of a gravitational field and latent heat flux to the approach by Herbert et al. [2013] and show that the resulting vertical temperature profiles are closer to observations compared with the previous approach for tropical conditions (e.g. Fig. 3c), but not for sub-arctic conditions (e.g. Fig. 4c). The authors also investigate the effect of ozone and different $CO2$ concentrations on the resulting vertical profiles of temperature and energy fluxes, but I was neither able to verify how these effects were implemented nor assess in how far they resemble observed effects. Table 1 suggests that the climate sensitivities to $CO2$ in all simulations was quite different to those presented in another paper, based on an Earth System Model. The authors explain some of the discrepancies between simulation results and observations by various simplifying assumptions in their model.*

**Authors response 2:** We will discuss the computation of the radiative budget below and we focus on the sensitivities to $CO_2$ computed in table 1. The goal of the paper is to verify if a more detailed account of the energy content of the atmosphere and of the energy fluxes helps to improve MEP based model of convection. Sensitivities computed for fixed absolute moisture are far away from those computed with an Earth System Model (ESM). However, the results show that for fixed relative moisture, and for tropical latitudes (the most pertinent for our model for reasons mentioned in the manuscript), the sensitivity of our model is closer to the ESM than the previous MEP based model of [Herbert et al., 2013] (fourth column of the table). In fact, sensitivities of the ESM correspond to a global response. So the reported values in the sixth column are presented only for qualitative comparison.

**2 General assessement and recommendation**

*The linguistic shortcomings, already pointed out by Referee 1, were a bit distracting and so plentiful that I gave up highlighting them, in order to focus more on the content. If this manuscript was going to be accepted for publication, it would need a serious revision for language and typography.*

**Authors response 3:** We want to apologize for linguistic and typographical errors (see comment to the first referee). We will give special attention to this point during the correction of the manuscript.

*The promise of the MEP principle to reduce the need for empirical parametrization and/or model calibration is huge and hence I generally welcome attempts to build MEP-based models and evaluate them rigorously. However, the work presented here left me puzzled in many places about its thermodynamic basis. As in previous work, the authors formulate entropy production as an energy flux between two layers divided by the temperature of one of the layers. However, here, the energy flux comprises not only sensible heat, but also water vapour and potential energy. One of the basic textbook expressions of entropy production is that it relates to the rate by which a given thermodynamic potential gradient is depleted, e.g. temperature, pressure, or chemical potential, in some treatments also gravitational potential [e.g. Kondepudi and Prigogine, 1998, Eq. 10.1.9]. This is expressed as a product of a thermodynamic force and a thermodynamic flow, e.g. by a reaction rate multiplied by the affinity of the reaction divided by temperature or a heat flux divided by the temperature difference of two systems. In the present paper, mass flux is multiplied by the sum of thermal, gravitational and latent energy (Eqs. 1, 2 and 5), but I cannot draw the connection of these equations to the relevant thermodynamic formulations. For example,*

*I do not see how the usual $TS$ term relates to $C_pT_i$ , as $S \neq C_p$ , or how $Lq_i$ is related to the classical $\mu N$ . I am also missing the $PV$ term in the formulation of total energy, and the consideration of volume work. Clearly, the authors should refer to established literature to help the reader reconcile their equations with thermodynamic theory.*

**Authors response 4:** The entropy production is here the product of affinity (gradient of inverse temperature which is the thermodynamical force that drives the flux) and energy flux $F = m\nabla e = m\nabla(C_pT + gz + Lq)$. More explicitly, the local entropy production is $\sigma = F\nabla(1/T)$. And it can be easily shown that the total entropy production (sum other all layers) can be rewritten as in equation 5. We consider only this term in the entropy production because we use the hydrostatic hypothesis (so there is no work of pressure force $PV$), and we don't consider the term due to chemical potential $\mu N$. This is usually the only term retained in MEP based model [Kleidon, 2010]. The terms $gz$ and $Lq$ are not affinities-fluxes product terms of the entropy production ($\mu N$) but nonthermal contributions to the energy flux. We will describe these standard atmospheric assumptions in more details in the revised manuscript.

*I am also very confused about the combination of energy balance (Eq. 3) and the radiative budget (Eq. 4), as it suggests that all shortwave radiation reaching a given layer is absorbed and converted to either longwave radiation or one of the other energy terms. I suppose that $R_i$ should be the absorbed shortwave radiation, but then the equation still misses the absorbed longwave radiation, which is the main reason for the greenhouse effect.*

**Authors response 5:** We are sorry for this misunderstanding that may originate from our notations, $\mathcal{R}_i$ represents the net radiative energy input in layer $i$ taking into account several effects: shortwave, longwave, reflexion, and reabsorption. More explicitly:

$$\mathcal{R}_i = SW_i + LW_i = SW_{i\downarrow} - SW_{i\uparrow} + LW_{i\downarrow} - LW_{i\uparrow}$$

where
$SW_{i\downarrow}$ is the downward radiative energy flux for shortwaves;
$SW_{i\uparrow}$ is the upward radiative energy flux for shortwaves;
$LW_{i\downarrow}$ is the downward radiative energy flux for longwaves;
$LW_{i\uparrow}$ is the upward radiative energy flux for longwaves.

This precise definition of $\mathcal{R}_i$ will be given earlier in the text. We will also give equation 4 before equation 2. If we consider that radiative heating is the only forcing of convection, and given the definition of $\mathcal{R}_i$, equation 3 is correct.

*The paper does not specify at all how $SW_i$ and $LW_i$ are calculated.*

**Authors response 6:** The radiative code was developed by [Herbert et al., 2013]. The detailed description is long so we referred to this original paper, and its supplementary material, for details. But your remarks push us to give more details about the radiative code in the future version of the manuscript. We will add the following precisions:
As specified by [Herbert et al., 2013]: "The purpose of this model is to reach a balance with a realistic description of the absorption properties of the major radiatively active constituents of the terrestrial atmosphere while keeping a relatively smooth dependence of the radiative flux with respect to the temperature profile. This last requirement is necessary to use the model in the framework of a variational problem.";
Computation of $LW_i$: In the longwave domain, the code decomposes the spectrum into 22 narrow bands, and in each band, it accounts for absorption by water vapor and carbon dioxide only. The absorption coefficient is computed using the statistical model of [Goody, 1952] with the data from [Rodgers and Walshaw, 1966]. For the spatial integration, the diffusive approximation is performed with the standard diffusion factor $\mu = 1/1.66$. Apart from the absorption data, given once and for all, the inputs of the model are the water vapor density and temperature profiles and carbon dioxide concentration. One may either fix absolute or relative humidity;
Computation of $SW_i$: In the shortwave domain, absorption by water vapor and ozone is accounted for by adapting the parameterization from [Lacis and Hansen, 1974]. The input parameters for the model are the water vapor density and ozone density profiles, as well as surface albedo and solar constant.
The radiative budget of the atmospheric layer i, $\mathcal{R}_i$, is given by summing over all terms involving the layer in question. In particular, $\mathcal{R}_i$ is a function of all temperatures $\{T_j\}_{j=0,...,N}$ in the profile

*I find even more puzzling the assumption that $m_i \geq 0$ (P6L1), termed as the "mechanical constraint". On P4L4, the authors define m i as either "the upward mass flux leaving the layer $i-1$" or "the downward mass flux coming to the layer $i-1$". This does not make any sense to me, as according to this wording, both would refer to the flux across the upper boundary of Layer $i-1$ but would have opposite signs. Whatever the correct*

*definition of $m_i$ , I do not see how setting it to a positive value represents a mechanical constraint and how this is reconcilable with the statement on P11L14 about a "downward convective energy flux" in the case of light absorption by ozone at the top of the atmosphere.*

**Authors response 7:** In our model, the definition of the mass fluxes comes from graph theory: for an oriented graph, there can be a link in both directions for edges. $m_i^+$ and $m_i^-$ are positive. $m_i^+ - m_i^-$ represents the net mass flux, while the minimum represents a mixing of mass. In the stationary state, the net mass transport must be 0 ($m_i^+ = m_i^- \equiv m_i$), but not necessarily the mixing. $m_i$ can be viewed as a diffusion coefficient. We need to impose this coefficient to be positive (i.e $m_i > 0$). Consequently, the energy flux is oriented from hot layer to cold layer. We have used the term mass flux for $m_i$, but it represents a mixing of mass. We will use the term mixing in the next version of the manuscript which is less confusing. We will also remove the unnecessary definitions of $m_i^+$ and $m_i^-$ .

*Last but not least, the article lacks a data availability statement. See www.earth-system-dynamics.net/about/data policy.html. Without the code and data used to generate the results and given the above-mentioned inclarities I am afraid that the manuscript is of very limited use.*

**Authors response 8:** The code used to compute profiles is a small part of a much larger code. It is likely to be of very limited value without documentation. The implementation of the code is not really important to describe the results. In fact, one can use any algorithm that solves the optimization problem defined in equation 8.

**2.1 Recommendation**
*While my above comments seem rather negative, I still believe that the paper could be a valuable contribution to the scientific literature, if the authors drew clear connections between their formulations and classical thermodynamic formulations, explaining all simplifying assumptions and if they provided all the information needed to reproduce their results. More rigorous comparison of simulations with observations would also be helpful. Since this, along with the necessary clean-up for language and typography, would require considerable effort, I would recommend a major revision or a recommendation for re-submission.*

**Authors response 9:** Our model does not aim at simulating precisely an atmospheric column since our assumptions are far too crude for that. The comparison with observations can only be qualitative.

**3 Specific comments**
*Eq. 4: What about incoming longwave, reflection and re-absorption? Why does SW depend on T? What about LW emissivity?*

**Authors response 10:** These effects are taken into account in the radiative model (see authors responses 5 and 6). However, $SW$ does not depend on temperature. We will correct this error in equation 4. We thank the referee for noting this mistake.

*P5L8: What does "non local dependence" mean?*

**Authors response 11:** The net radiative budget of the atmospheric layer i, $\mathcal{R}_i$, naturally depends on every temperatures $\{T_j\}_{j=0,...,N}$ and greenhouse gases concentrations because of reabsorption and reemission and not of characteristics of layer $i$ alone. We hope that a more detailed description of how the radiative budget is computed will help the reader (see authors response 6).

*P5L21: This does not make sense, as mass flux can go in both directions.*

**Authors response 12:** $m$ correspond to a mixing coefficient that should be positive and not to mass flux. We will change the definition in the next version of the manuscript as explained above (see authors response 7).

*P6L14: What does "absolute ratio moisture at saturation" mean? What are the units of $q_s$?*

**Authors response 13:** $q$ is the mixing ratio (the ratio between the mass of water vapor and the total mass of the air for a given volume) which is a dimensionless quantity. And $q_s$ is the value of $q$ for the saturated air. We will replace "absolute ratio moisture" with "mixing ratio", and give the definitions of $q$ and $q_s$ in the text.

*P6L15: How is pressure prescribed? How does it vary with height?*

**Authors response 14:** For atmospherical modeling, it is common to work with pressure as vertical coordinates. Then, the elevation $z$ depends on the temperature profile. The computation of geopotential $gz$ is detailed in Appendix B. We will state explicitly that pressure levels are prescribed in the legend of figure 1.

*P7L10: What is the thermodynamic force driving the flux? Please explain why flux can be in the opposite direction to the energy gradient.*

**Authors response 15:** As said above, the thermodynamic force driving the flux is the usual gradient of inverse temperature. In previous MEP based model like [Herbert et al., 2013], nothing forbid the flux to be opposed to the energy gradient at some point if it allows to attain a larger entropy production state. In our model, the constraint (positivity of $m_i$) impose the direction of the flux everywhere (see authors response 7).

*Fig. 3: Is this for the tropical latitudes? Please specify what the data in each figure corresponds to and how it was generated.*

**Authors response 16:** This figure is the results of the model for tropical latitudes. All the results come from the numerical resolution of the optimization problem described in AppendixA. We will be more explicit on this point in the figure legend.

*P10L26: Why would insolation vary more at high latitudes? It would vary more at the seasonal scale, but less at the diurnal scale. Which variation is more important for the profiles simulated here?*

**Authors response 17:** Since we use measurements [A. McClatchey et al., 1971] that depend on season and latitude, we are talking about the seasonal variations here. We will be more explicit in the next version of the manuscript.

*P11L1-5: I do not understand the first sentence at all and the whole paragraph appears rather hand-waving to me. Why does the current version give worse results for high latitudes than Herbert et al. [2013]? More analysis and explanation would be necessary to generate any definitive insights here.*

**Authors response 18:** In this paragraph, we try to explain why our model is not realistic for arctic regions. We don't compare results to [Herbert et al., 2013]. We just want to insist on the point that a single column model without lateral fluxes is not very appropriate for polar conditions. We will try to reformulate better this paragraph.

*Table 1: Where the literature results based on the same albedo values? How were these values chosen?*

**Authors response 19:** We have chosen representative (typical) values of albedo. They don't come from a precise article. Again, the comparison with given reference profiles is for illustrative purpose, not for a precise evaluation of the model. We will describe these albedo values in the text as "typical".

*P13L1-2: This sentence is not clear. Was it a discretization effect or not?*

**Authors response 20:** We have used the model of [Herbert et al., 2013] to compute the sensitivities of the fourth column of table 1. However, these values are different from those of the original article. We have verified that it was only due to the fact that [Herbert et al., 2013] have used $N = 9$ layers but we choose $N = 20$. We will erase the citation in the table and give more precise explanation in order to avoid any ambiguity.

*P15L29: Observation-based temperature profiles were presented, but what are realistic "energy content, and energy fluxes profiles"?*

**Authors response 21:** Realistic energy and energy fluxes profiles referred to the global structure that described a stratification (which are not obtained by previous MEP based model, up to our knowledge). Measurements energy fluxes and energy content are not given in [A. McClatchey et al., 1971]. We will replace "realistic" with "physically relevant".

*References: Why are the references not sorted by author name?*

**Authors response 22:** We will use bibtex correctly in the revised manuscript.

**References**

[A. McClatchey et al., 1971] A. McClatchey, R., W. Fenn, R., E. Volz, F., and S. Garing, J. (1971). Optical properties of the atmosphere (revised). *Environ. Res. Pap.*, 411:100.

[Goody, 1952] Goody, R. M. (1952). A statistical model for water-vapour absorption. *Quarterly Journal of the Royal Meteorological Society*, 78(336):165–169.

[Herbert et al., 2013] Herbert, C., Paillard, D., and Dubrulle, B. (2013). Vertical temperature profiles at maximum entropy production with a net exchange radiative formulation. *Journal of Climate*, 26(21):8545–8555.

[Kleidon, 2010] Kleidon, A. (2010). A basic introduction to the thermodynamics of the earth system far from equilibrium and maximum entropy production. *Philosophical Transactions of the Royal Society B: Biological Sciences*, 365(1545):1303–1315.

[Lacis and Hansen, 1974] Lacis, A. A. and Hansen, J. (1974). A parameterization for the absorption of solar radiation in the earth's atmosphere. *Journal of the Atmospheric Sciences*, 31(1):118–133.

[Rodgers and Walshaw, 1966] Rodgers, C. D. and Walshaw, C. D. (1966). The computation of infra-red cooling rate in planetary atmospheres. *Quarterly Journal of the Royal Meteorological Society*, 92(391):67–92.

---

## Author Response (AR2)

**Responses to the comments of Stan Schymanski and the marked-up version of the manuscript**

The authors have improved the manuscript considerably, and I would like to thank them for the helpful responses to my comments. Unfortunately, some of my major concerns remain:

Authors response 1: We thank Stan Schymanski for his useful comments. His major concern (point 1 below) arises from a misunderstanding of our modeling procedure that was not explicit enough in the previous version of the manuscript. We are now giving a much more detailed description of our convective model.

1) The computation of entropy production is not consistent with the thermodynamic literature: The authors suggest that the difference in inverse temperatures drives the exchange of "moist static energy", consisting of sensible heat, latent heat and potential energy (Eq. 1), and therefore the relevant entropy production is the between-layers exchange of moist static energy divided by temperature (Eq. 5). However, the components of "moist static energy" exchange are actually driven by different forces. Only the sensible heat component is driven by a temperature difference, whereas latent heat flux is ultimately driven by a vapour concentration difference and potential energy changes are driven by buoyancy differences. Hence the entropy production terms associated with each of these components should be calculated separately.

Authors response 2: The way we compute the entropy production is indeed different from some standard expressions given in the literature where the entropy production mainly concerns the diffusive processes, in which, as correctly underlined by the reviewer, the flux of different quantities are driven by different thermodynamic forces. But we aim here at representing convection, not diffusion. Fluxes are consequently not independent or "driven by different thermodynamic forces". In contrast, fluxes are all associated with the same mass transport. Air parcels are mixed only after (turbulent) convective motions. We can separate this process into two steps:

1. We assume a pseudo-adiabatic motion from one box to another of an air parcel due to convection. This step is purely mechanical and without any entropy production since we neglect the viscous dissipation. The energy of the air parcel is conserved, but its temperature and composition may change during the motion. If the air parcel was initially with temperature  $T_i$ , at elevation  $z_i$  and humidity  $q_i$  and goes at elevation  $z_{i+1}$  where the humidity is  $q_{i+1}$  and temperature  $T_{i+1}$ , the temperature of the air parcel becomes

$$T'_{i} = T_{i} + \frac{1}{C_{p}} \left[ g(z_{i} - z_{i+1}) + L(q_{i} - q_{i+1}) \right].$$
(1)

2. Then, the air parcel of mass m is mixed by diffusion with the ambient air at elevation  $z_{i+1}$  and transfers an amount of sensible heat  $mC_pT'_i$ . At the same time, an air parcel of mass m leaves the layer at elevation  $z_{i+1}$  with sensible heat  $C_pT_{i+1}$ . The convergence of sensible heat fluxes due to this process is, therefore

$$mC_p(T'_i - T_{i+1}) = m(e_i - e_{i+1}),$$
(2)

$$e = C_p T + gz + Lq. aga{3}$$

So the entropy production resulting from the sensible heat exchange between layers i and i + 1 is the product of the flux  $m_{i+1}(e_i - e_{i+1})$  with the thermodynamic force associated to the sensible heat only (i.e. the gradient of inverse temperature)

$$\sigma_{i+1} = m_{i+1}(e_i - e_{i+1}) \left(\frac{1}{T_{i+1}} - \frac{1}{T_i}\right) \equiv F_{i+1} \left(\frac{1}{T_{i+1}} - \frac{1}{T_i}\right).$$
(4)

where  $m_{i+1}$  is the mass mixing coefficient between layers i and i+1.

where

Our definition of entropy production is therefore fully consistent with the literature. The other terms that usually appear in the entropy production (mixing entropy of water vapour, viscous dissipation of mass transport, ...) are not taken into account. These steps were not explicitly detailed in the manuscript. Therefore, we have added the following lines (page 5 line 8 to page 6 line 20)) to explain this point in the revised version of the manuscript:

"In the thermodynamics of diffusive processes, the entropy production is expressed as the sum of products of the fluxes with their associated thermodynamic forces. But we aim here at representing convection, not diffusion. We need to represent how the air parcels are mixed but only after (turbulent) convective motions. We can usefully separate this process in two steps. First, we assume a pseudo-adiabatic motion of an air parcel from one layer to another due to convection. This step is purely mechanical, without entropy production since we neglect viscous dissipation. The energy of the air parcel is conserved, but its temperature and composition may change during the motion. If the air parcel was initially in layer i and goes to the layer i + 1, the temperature of the air parcel becomes, by conservation of energy,

$$T'_{i} = T_{i} + \frac{1}{C_{p}} \left[ g(z_{i} - z_{i+1}) + L(q_{i} - q_{i+1}) \right].$$
(5)

Here, we have assumed that the water vapour concentration changes pseudo-adiabatically during the convection and not due to the mixing. This is not fully consistent since we do not impose water conservation. Secondly, the air parcel is mixed by diffusion with the ambient air in layer i + 1 and transfers an amount of sensible heat per unit mass  $C_pT'_i$ . At the same time, air parcels leave the layer i + 1 for the layer i with a sensible heat per unit mass  $C_pT_{i+1}$ . So the net flux of sensible heat due to this process is

$$m_{i+1}C_p(T'_i - T_{i+1}) = m_{i+1}(e_i - e_{i+1}) = F_{i+1},$$
(6)

where  $m_{i+1}$  is the mass mixing coefficient between layers i and i + 1. So the entropy production that results due to the sensible heat exchange between layers i and i + 1 is the product of the flux  $F_{i+1}$  with the thermodynamic force associated to the sensible heat only (i.e. the gradient of inverse temperature)

$$\sigma_{i+1} = F_{i+1} \left( \frac{1}{T_{i+1}} - \frac{1}{T_i} \right).$$
(7)

By summing over all layers, and using the fact that  $F_{N+1} = 0$ , we show that the total entropy production can be written

$$\sigma = \sum_{i=0}^{N} \frac{(F_i - F_{i+1})}{T_i}.$$
(8)

In thermodynamics, more terms may contribute to entropy production such as volume work or mixing. As for others MEPM [Kleidon, 2010], we only retain the sensible heat exchange term. The geopotential and latent heat terms appear in the entropy production only as a result of our representation of convective transport, which is supposed to occur as a mechanically induced mass transport without entropy production."

2) Disregard for the second law of thermodynamics at local scale: In Eq. 5, the authors formulate the total entropy production as the sum of entropy production terms related to each between-layer exchange. On P6L10, the authors then formulate  $m_i > 0$  as an additional constraint. This would not be necessary if they actually respected the second law of thermodynamics for each inter-layer exchange individually, as negative mi would result in negative entropy production terms (this is why de-mixing does not happen spontaneously).

Authors response 3: The constraint  $m_i > 0$  does indeed imply the second law at local scale, and the local entropy production would be negative if we would not impose  $m_i > 0$  as explained at p-8 lines 13-22. Nevertheless, we prefer the discussion in terms of mass transport, described in section 3.1, which gives a mechanical description of how the energy is transported (i.e. the mechanics). Such an interpretation allows us to represent explicitly the transport of moist static energy. We have mentioned it in the revised version of the manuscript page 11 line 4:

"As a consequence, the local entropy production is also constrained to be positive as in [Ozawa and Ohmura, 1997]. But with an explicit account for the mass transport as explained above, we can account not only for sensible heat, but more generally for moist static energy transfers in a convective column."

3) The analysis and results are not easily reproducible, if at all. I agree that a large code without documentation is of very limited use, but I think that a paper without code and data is of even less use. I cannot see how Figures 3-6 could be reproduced using the information given in this manuscript. The data availability statement is still missing.

Authors response 4: We had in fact supplied a Python code to reproduce the results (temperature, energy, and flux profiles) available here https://doi.org/10.5281/zenodo.2597543. We specify the link in the section code availability before appendices in the revised manuscript. The code is not the one that was used to derive the results presented in the manuscript, but we have checked that the output of the Python code are identical to those in the manuscript. The code uses the module scipy.optimize which is the standard toolbox for optimization in Python. The modifications in the manuscript are page 17 line 11:

[revised manuscript text omitted]

---

## Author Response (AR3)

**Final version of the manuscript**

Dear Handling Editor,

Please find all the documents necessary to reproduce the final version of the manuscript in Latex. We have replaced "in term of" by "in terms of", sometimes replaced by similar expressions ("as a function of" or "in matters of") in order to avoid repetitions.

Best regards,

Vincent Labarre, Didier Paillard and Bérengère Dubrulle.